# Fibration symmetries and cluster synchronization in the *Caenorhabditis elegans* connectome

**Bryant Avila**[1☯], **Matteo Serafino**[1], **Pedro Augusto**[2,4☯], **Manuel Zimmer**[3,4‡], **Hernán A. Makse**[1‡]*

**1** Physics Department, Levich Institute, City College of New York, New York, NY, United Stated of America, **2** Vienna Biocenter PhD Program, Doctoral School of the University of Vienna and Medical University of Vienna, Vienna, Austria, **3** Research Institute of Molecular Pathology (IMP), Vienna Biocenter (VBC), University of Vienna, Vienna, Austria, **4** Department of Neuroscience and Developmental Biology, Vienna Biocenter (VBC), University of Vienna, Vienna, Austria

☯ These authors contributed equally to this work.
‡ MZ and HAM also contributed equally to this work.
* hmakse@ccny.cuny.edu

**Data Availability Statement:** A GitHub repository containing a user-friendly (quick to install) Matlab app capable of reproducing all of the simulations carried out in this paper can be found at: https://github.com/makselab/C.-elegans_locomotive_

## Abstract

Capturing how the *Caenorhabditis elegans* connectome structure gives rise to its neuron functionality remains unclear. It is through fiber symmetries found in its neuronal connectivity that synchronization of a group of neurons can be determined. To understand these we investigate graph symmetries and search for such in the symmetrized versions of the forward and backward locomotive sub-networks of the *Caenorhabditi elegans* worm neuron network. The use of ordinarily differential equations simulations admissible to these graphs are used to validate the predictions of these fiber symmetries and are compared to the more restrictive orbit symmetries. Additionally fibration symmetries are used to decompose these graphs into their fundamental building blocks which reveal units formed by nested loops or multilayered fibers. It is found that fiber symmetries of the connectome can accurately predict neuronal synchronization even under not idealized connectivity as long as the dynamics are within stable regimes of simulations.

## Introduction

Advances in reconstructing synapse resolution wiring diagrams of various model organisms [1–6], demand the development of new computational tools that make predictions of how neuronal wiring relates to neuronal function [7]. The nematode *C. elegans* is an ideal system to prototype such approaches because of its fully mapped and well characterized small nervous system of just 302 neurons [1, 8–10]. Various studies have identified structure in the wiring architecture of the worm connectome e.g., over-represented network motifs [8, 11], small worldness [12], rich club topology [13] as well as community structure and functional layers [9, 14–16]. While such features suggest functional implications such as sensory-motor flow or wiring economy, they typically fall short of making concrete predictions for how neurons dynamically interact with each other.

simulator (MIT license). The only additional Matlab toolboxes besides the basic ones in Matlab R2021b that are needed are the Statistic and Machine Learning toolbox and the Fuzzy Logic toolbox. This Matlab app also contains tools not mentioned here which can be used to further explore synchronization in the networks studied in our manuscript. Within this repository there is a (Sage 9.3) python notebook used to find orbit partitions, additionally this repository also contains all files needed to reproduce all of the networks explored in this paper. To reproduce the fibration partitionings found in our paper we implemented a code that can do such which can be found here: https://github.com/makselab/fibrationSymmetries.

**Funding:** Funding leading to the results of this work was provided to both H.M. and M.Z. by The National Institute of Biomedical Imaging and Bioengineering and National Institute of Mental Health through 1023 the National Institute of Health BRAIN Initiative Grant R01 EB028157. https://www.nibib.nih.gov/ https://www.nimh.nih.gov/ https://braininitiative.nih.gov/ M.Z. is supported by the Simons 1024 Foundation (543069). The Research Institute of Molecular Pathology is funded by Boehringer Ingelheim. https://www.simonsfoundation.org/ https://www.boehringer-ingelheim.com/ The funders had no role in study design, data collection and analysis, decision to publish, or preparation of the manuscript.

**Competing interests:** The authors have declared that no competing interests exist.

One notable type of dynamics observed in *C. elegans* is the brain-wide synchronization of neural activity across well-defined ensemble of neurons; where a general notion of synchronization is that larger ensembles of neurons have correlated activity patterns [17–19]. Understanding how such synchronizations are supported by the underlying connectome is a major challenge in neuroscience and can provide mechanistic insights into how the brain processes information. We recently showed that primary and secondary input similarities predict pairwise synchronizations in *C. elegans* neuronal dynamics [17] hinting at symmetries in the larger network context underlying such synchronisations. A mathematically precise description of such input similarities was missing with which symmetries could be uncovered. Therefore, it is important to better define the structures in the connectome that lead to the synchronization of groups of neurons [20, 21] based on their inputs.

Synchronization is ubiquitous across all organisms and at different scales. Some examples are gene co-expression patterns [22], metabolism, hormonal regulation, cell communication, and cardiac muscle. As such, the results on this direction may be generalized to the different levels of synchronization in biological networks.

From a theoretical perspective, it is well known that the symmetries of the underlying network, including automorphisms (orbits) [23–25] and fibrations (fibers) [26–30], can strongly determine the dynamics of the system leading to synchronization of neurons within clusters [31–33], i.e. cluster synchronization. Orbits are related to particular permutations (automorphism) of the network that preserve the network's adjacency connectivity structure- including both in-degree and out-degree. The cluster of nodes subject to these permutations are called orbits, and nodes in orbits synchronize their activity under a suitable dynamical system that is admissible with the network [23, 29, 33]. The nodes in any of these clusters synchronize due to receiving equal amounts of input from the same or equivalent sets of nodes. This procedure can go on ad infinitum, meaning to be extended for as many existing network nodes. This synchronization is explained in more detail further in the paper.

The requirements for the existence of orbits are strict, as they must preserve the entire adjacency matrix network structure. Synchronization of neuronal activity patterns, on the other hand, only requires constraints on the nodes' inputs (in-degree), not the outputs (out-degree).

Fibrations are symmetries that generate automorphisms only requiring the invariance of the input tree structure of each node [27, 28, 32, 34–36]. These symmetries form symmetry groupoids rather than symmetry groups of automorphisms where these define fibers similarly to orbits, where nodes in the same fiber synchronize [37]. In this sense, a groupoid is a more general algebraic structure than a group, where a groupoid need not be associative; by such fibers are more general than orbits, as they are related to equivalent classes which are referred to in the math community as *balanced coloring*. Every orbit is a fiber, but not every fiber is an orbit. Therefore fibers capture more patterns of cluster synchronization than orbits. Therefore, we here propose that fibers are a more general and mathematical defined description of connectivity that relate to the input similarities investigated in ref. [17]. In [37] we have found automorphisms in the structure of the *C. elegans* connectome, specifically in the gap junction and chemical synapses networks of motor circuits involved in forward and backward locomotion. Here we generalize this study to search for more general patterns of symmetry fibrations and theirs associated fibers of cluster synchronization in the *C. elegans* locomotive connectome.

Finding perfect symmetries in any biological network is highly unlikely, therefore we expect biological networks to exhibit more fibrations symmetries than orbit symmetries, due to the milder constraint of fibers as compared to orbits, (if any symmetries are detected in the first place). In [26], we have found that gene regulatory networks of many organisms spanning from simple bacteria to humans are composed of fibration symmetries. These symmetries

partition the network into synchronized fibers which are then the building units characterizing the structure of such networks. As such, fibrations can be thought of as the "building blocks" of a network, in the sense that they represent the fundamental units or components that can be combined to form larger and more complex structures or systems. According to [38], the building blocks of a network can be thought of as the "units of computation" in a neural network, similar to how transistors constitute the basic building blocks of electronic circuits.

The concept of building blocks as a way to modularly construct neural networks with specific desired properties is familiar in the field. In [39–41] a framework has been proposed for understanding the function and organization of neural networks in terms of building blocks known as "network motifs." Network motifs are small, recurring patterns of connections within a network that are thought to perform specific functions or play a role in shaping the network's overall structure and function.

Symmetries in the connectome can only tell us about the existence of synchronized solutions but not about the stability of such solutions. If a group of neurons were prepared to be in a synchronous state, theory predicts that these will stay in such a state indefinitely if no noise is present [25, 42]. However, if the system were to start in any other initial condition, would it evolve towards a synchronized state, such as a fixed point? If so, the synchronous state is considered to be stable. If not, the synchronous state is considered to be unstable, such that most, if not all, neurons have different values at any given time (asynchronous). The structure of the connectome alone cannot predict their stability. Stability also depends on the particular system of equations that define the neural dynamics of the system. It is not a property that is inherent to the fibration symmetry itself but rather a property that emerges from the interactions between fibers and their surroundings through the dynamical system. Thus, a single symmetric connectome leads to synchronous solutions, but these solutions could have different stability properties according to different dynamical systems of equations. Since the same connectome with a different dynamic system may lead to a stable or unstable solution, stability needs to be investigated for each particular dynamical system of equations [25, 42].

In this paper, we explore the structure-function relationship of a version of the neuron network in *C. elegans* used in [26] related to the locomotion function (as described later) to characterize its fibration symmetries and its comparison with automorphisms. Generalizing the results found in [37] on the existence of automorphisms, here we further studied the more general fibrations, their associated fiber partitions, and their relations with the orbital partitions obtained from automorphisms. We apply a set of simplifications to make our theoretical and modeling approaches tractable and which are prerequisite to identify the orbits and fibers in this study in a mathematically concise manner. First, we focus on sub-networks that have been assigned to distinct behavioral functions of C. elegans. Next, we study the graphs of chemical synapse and gap junction networks. It is true that any nervous system relies on both gap junctions and chemical synapses. However, it is known that both types of connections contribute differently to the overall activity of a given circuit. For example, gap junctions are known to be more important for the transition between the backward and forward circuits rather than the generation/propagation of these behaviors separately [43]. Moreover, specifically in the backward circuit, gap junction removal does not disrupt A type motor neuron synchronicity, but reduces the strength of these motor neurons. This happens because the interneuron AVA activates A type motor neurons through acetyl cholinergic signaling and is the major driver of this reverse locomotion [44]. Additionally, as will be seen, simulations of a network of neurons solely composed of gap junction connections with no driving input stimuli will settle into a global synchronous state, meanwhile a network composed solely by chemical synapses and no external stimuli will settle into different clusters fully dependent on the

network's symmetries Altogether, despite agreeing that studying these connections in separate might not be ideal, there is previous experimental evidence for their individual contribution.

Moreover, we apply subtle modifications (repairs) since the mathematical procedures applied here are strict and do not account for variability in connectome data and even slight deviations from perfect symmetries. Lastly, we use simplified neuronal models that treat each network node as if they had identical dynamical properties. These models solely serve as a test case to confirm that our predictions about cluster synchronizations hold in the context of a dynamical system, in contrast to a sole static neuronal connectome. Below, we will discuss the implications of our findings, made under these simplifications, for the worms neuronal network functions and behaviors it produces. We compare the synchronous solutions obtained from a dynamical system of interacting neuron equations with the ones predicted by the fibers of the connectome and investigate their stability under different initial conditions. Simulating the dynamics of neurons is achieved by mapping admissible in-degree dependent coupled ordinary differential equations (ODEs), as explained further in the paper. We characterize the locomotion connectome into fiber building blocks organized into classes according to some of their structural properties, such as, how many inputs these have and a characterization of how many closed paths these inputs form with the fiber. These building blocks are similar to those found in genetic networks [26] implying the generality of these blocks for the structure of biological networks.

We follow two steps. First, we partition the neuron network using fibrations and automorphisms. As anticipated, we find that biological networks tend to have more fibers than orbits when examining a directed network of chemical synapses. In the network of gap junctions, the fibers are the same as the orbits as expected for an undirected network. Next, we investigate the stability of the synchronous solutions predicted by the symmetries when the network is exposed to different external stimuli using ordinary differential equations (ODEs). Initially, we observe that without an external driving force the ODEs settle into the same synchronization groups as predicted by symmetries and expected by theory. Then we examine the synchronizations that occur in the network when it is driven by an external driving input; its observed that these stimulated neurons react in a stable fashion similarly like a set of neurons in electrophysiological studies when stimulated with an external current source [45, 46]. We find that instabilities can appear above certain magnitudes for the external input which are model dependent. We then simulate the case where the network's weight edges are not fully known and study the impact of this missing information on the synchronization states. We found that the effects of varying weights impact the synchronicities of the networks. The robustness to change varies among the different types of networks, with the Gap networks being more robust to change.

We conclude our analysis by focusing on the partition of fibers, which serve as building blocks for each of the networks studied here. These can be topologically categorized, information that in turn can be used to determine the relative stability against perturbations for each fiber. The symmetries structures and the synchrony patterns we find here could be tested in in-vivo in the future by opto-genetically silencing or ablating neurons which would affect the synchronization predicted by fibers. [47, 48]. Some building blocks with different structure can belong to the same topological classification where their functionality could be tested by analyzing the similarity of the dynamics of the neurons belonging to the same fiber.

The paper is structured as follows. The locomotion network section for the latter portion of the Introduction and focuses on the construction of the neurons' network and the data set used in this paper. Then in Fibers and orbits the theory used in this paper is introduced which covers equitable partitioning, fibers, orbits and the methods used for our analysis, followed by the section Building blocks in which fibers are used to construct elementary sub-networks

which form the locomotion network. Ending the theory presentation with Admissible ordinary differential equations that outlines the admissible ODEs used to conduct simulation tests and with Synchronicity measure. After the theory we move into the results with first showing in Network partitions in the locomotion connectome how the locomotion networks of the *C. elegans* are partitioned by fibers and orbits. Additionally the fiber building blocks of the directed networks are shown. Results are concluded in Network simulations with the synchronicities found from numerical methods solving ODEs applicable to the worm neuron system. Wrapping up with the sections of Discussion and conclusion Conclusion.

## The locomotion network

A comprehensive analysis of orbits across the entire connectome is currently out of reach, due to computational constraints. Therefore, we focus on two locomotion sub-networks implicated in the generation of two distinct gaits: forward and backward behaviour. One major reason for this selection is the fact that the C. elegans locomotion network has feedfoward excitatory synapses from the command inter-neurons to their respective motor-neurons, inducing a highly synchronous state in all neurons which participate in the motor output. In addition, secRNA seq data has shown that motor-neurons represent the most similar class of neurons in the worm, which further supports the comparisons made later in this study [49]. The networks present in this paper are manually repaired version of the Varshney connectome (available at the WormAtlas website [50]). Such repaired networks have been validated by integer programming repair algorithms, one based on full fibration analysis [51] and another on quasi-fibrations [35]. The former leads to the network solutions found in [37] and studied here when provided with a set of groups of neurons belonging to each fiber. The reason behind such repairs is to bring to the fort front the rich number of symmetries in the raw connectome with only a few modification to it. Where without modification each neuron exhibits its own unique symmetry, the slightly modified connectome captures multiple groups neurons belonging to one symmetry. The repairs were done by hand by removing and adding a minimum amount of edges which would reduce the number of fibrations in each graph by the largest amount possible given the restriction that the each change of the network was not outside natural variation of edges of 25% [8, 37], where most modification did not reach the 25% variation. The repairs allow us to focus on the neurons with evidence to be involved exclusively in this animal's forward or backward locomotion. It includes inter-neurons AVB, PVC, and RIB for the forward system, inter-neurons AIB, AVA, AVD, AVE, and RIM for the backward system [52–54] and four classes of motor-neurons: DA, VA for the forward system and DB, VB for the backward system [55]. The remaining motor neurons (AS, DD, and VD) have not been included in the studied networks. This strategy is based on previous experimental findings showing that AS neurons are not mutually exclusively involved in either forward or backward movement [56]; inhibitory GABAergic DD and VD motor neurons are not strictly required for movement, i.e. when removed or blocked, the animal moves indistinguishably at a lower frequency and does not produce higher-frequency undulations. Nevertheless, it does not eliminate the ability to move forward, or backward [57]. The exclusion of DD and VD motor neurons allows partitioning the neurons mentioned above into two not overlapping functional sets (forward and backward locomotion). This stratagem avoids neurons belonging to multiple synchronization modes, resulting in more interpretable results.

Neurons in C. elegance can be physically connected through chemical synapses or gap junctions. Chemical synapses are directed connections meaning that a chemical signal can only propagate in one direction (from neuron A to neuron B and not vice versa) [58]. Gap junctions, on the other hand, can lead to undirected connections, allowing rapid propagation of an

electrical signal between two connected neurons in both directions [59]. Henceforth, we apply this simplification and ignore the possibility of rectifying gap junctions [60]. As such, the neuron network in the *C. elegans* can be represented by two adjacency matrices, one for each type of connection, chemical synapses and gap junctions. To better comprehend these two types of connections and their co-occurrence, we redirect the reader to [8].

Removing the inter-connections between the two functional sets of neurons for the forward and backward locomotion leads to four independent adjacency matrices representing the *C. elegans* locomotion system [37]: Forward gap junction (F-Gap), backward gap junction (B-Gap), forward chemical synapses (F-Chem), and backward chemical synapses (B-Chem) networks. Note, that this partitioning into forward and backward groups is justified since activity of these sub-netwoks has been shown to be mutually exclusive and tightly associated with either one of the two distinct motor programs [61–63]. In the original work of [37], the networks and their respective adjacency matrices are binary, meaning the connections between neurons either have a weight of 1 for existing edges or 0 for non-existing edges. In this paper, we also introduce and analyze the integer-weighted version of these sub-networks, where weights have positive integer values reflecting the number of synaptic connections between neurons based on Varshney's connectome [8]. These weights respect the symmetrization done by [37] and the fiber partitionings corresponding to the binary versions. This leads to a total of 4 sub-networks each with two edge weight versions under study.

## Materials and methods

### Fibers and orbits

The neural networks governing the movement of *C. elegans* are represented as a graph $G = (N_G, E_G)$, where $N_G$ is the set of nodes (neurons) with $n = |N_G|$ being the number of nodes and $E_G$ is the set of connections among neurons. The set of connections $E_G$ can be further decomposed into two separate types representing chemical synapses $E_{Chem}$ and gap junctions $E_{Gap}$, where are a more formal definition of a locomotion circuit as a graph would be: $G = (N_G, E_{Chem}, E_{Gap})$. Chemical synapse networks consist of directed connections denoted as $e_G^{u \to v}$ or $e_{Chem}^{u,v}$ from neuron $u$ to neuron $v$, while gap junction networks consist of undirected connections denoted as $e_G^{u \leftrightarrow v}$ or $e_{Gap}^{u,v}$ between neurons $u$ and $v$. A general connection in $G$ would simply be denoted as $e_G$. A connection is represented as ordered pairs of nodes for directed connections $(u, v)$, where $v$ is the head node and $u$ is the tail node. We interchangeably refer to these two types of connections as edges through out the paper. For undirected connections, two directed connections in opposite directions are used. Each edge has a head node $h(e_G^{u \to v}) = v$ and a tail node $t(e_G^{u \to v}) = u$.

Partitioning methods detect clusters of nodes with shared properties in a network. Two standard methods for partitioning networks into synchronized groups of neurons are Fiber [26–30] and Orbit partitioning [23–25]. These methods partition the $n$ nodes of a graph $G$ into groups of nodes which synchronize under admissible ODE dynamics [30–32, 64]. Admissible ODEs ensure that each node in the network has one ODE which is coupled to other ODEs in the same way its representative node receives connections. If the initial states of the nodes are within the stable regime of their given set of ODEs, then the nodes belonging to a partition will eventually reach synchronicity [29, 65]. Before diving into these methods, we must introduce the concept of equitable partition.

Loosely speaking, partitioning the $n$ nodes of a graph $G$ into groups that are in-degree conserving is an "equitable partitioning"; these groups of nodes are referred to as *cluster cells*. More precisely, a partition $\pi$ of $G$ with cells $C_1, \ldots, C_k$ is equitable if the number of in-degree edges from $C_j$ onto a node $v \in C_i$ depends only on the choice of $C_i$ and $C_j$. Neurons in the

same cluster cell have the same *balanced coloring* [32]. The out-degree can also be considered, but the in-degree conservation alone is sufficient for synchronization, and is therefore used in this paper [24, 25]. Fiber and orbit partitioning are examples of balanced colored partitioning methods with distinct characteristics, which are explained below.

**Fibrations, input trees and the minimal balanced coloring.**    A fiber partitioning relies on the notion of neighboring nodes. We begin by defining the immediate in-degree neighboring nodes of a given node $v$ as the multi-set of nodes fulfilling the following condition

$$t(e_G^{1 \to v}) = [u : u \in N_G \wedge (u, v) \in E_G] \tag{1}$$

where the multi-set of edges connecting into node $v$ are given by

$$e_G^{1 \to v} = [(e_G, 1) : e_G \in E_G \wedge h(e_G) = v] \tag{2}$$

where $(e_G, i)$ is a pair given by a positive integer number $i$ indicating the length of a walk terminating in $v$ and an edge at the start of said walk. The definition of a walk, in graph theory terms, is a finite or infinite sequence of edges which joins a sequence of vertices such that these sequences can have edge and vertex repeats. A walk is said to be closed when the first vertex and last vertex of its sequence are the same. For completeness we define the set of walks with zero length culminating on node $v$ to be empty and the tail nodes to be composed solely by the node $v$ as described below

$$e_G^{0 \to v} = [\,], \; t(e_G^{0 \to v}) = [v] \tag{3}$$

Given the above the fiber partitioning is associated with the input history of a node, which includes the input edges it receives from its immediate neighboring nodes, as well as the input those nodes receive from their immediate neighboring nodes. This notion can be extended an infinite number of times, resulting in a (possibly infinite) *input tree* [23]. The input tree for a node $v$ is denoted as $\mathcal{T}(v)$, which is the complete set of all walks that terminate on node $v$, giving rise to a rooted tree graph [28]. Such a structure can be placed in mathematical terms as

$$
\begin{aligned}
\mathcal{T}(v) \quad &= \bigcup_{i=0}^{N} \bigcup_{m=t(e_G^{i \to v})} e_G^{i \to m} = \\[1em]
&\bigcup_{i=0}^{N} \bigcup_{m=t(e_G^{i \to v})} [(e_G, i) : e_G \in E_G \wedge h(e_G) = m].
\end{aligned}
\tag{4}
$$

Where N represents the largest length to be considered for the set of walks of length N culminating in $v$ where for each length there is a layer in the input tree. As an example the first layer of the input tree of a node $v$ is composed of all the immediate in-degree neighboring nodes of $v$ plus their out-degree edges (connecting to $v$) which are all the possible walks of length 1 terminating in $v$.

An input tree can be represented visually, as shown in Fig 1, which illustrates the input history of nodes E and F in the network shown in Fig 2A. Each node in the network has its own input tree, which may appear different at first. However, if the labels of the nodes and edges are ignored, a symmetry can be uncovered, revealing that some nodes in the network have the same input tree structure. These equivalent structures are referred to as *isomorphic input trees*, which are bijective mappings from one input tree $\mathcal{T}$ to another input tree $\mathcal{U}$, such that there is a one-to-one mapping of nodes and edges from the $i$th layer of tree $\mathcal{T}$ to the $i$th layer of tree $\mathcal{U}$ [66]. As a note, $N$ can potentially be infinite if there are loops in the network (walks with node $v$ at the start and end of it); although the smallest length of walks $N$ to guarantee the complete

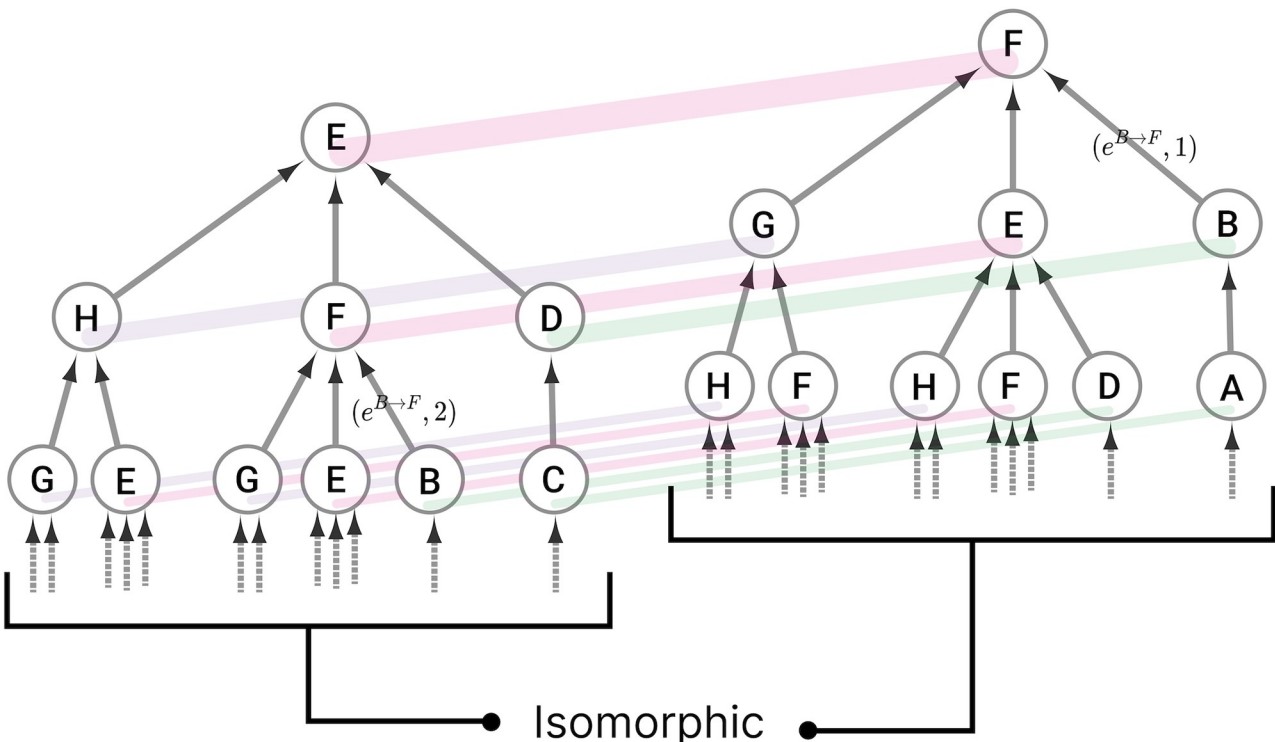

**Fig 1. Input trees and their isomorphism.** The input trees of nodes E and F for the network in Fig 2A are shown. The input tree of node E is given by all the possible paths in the network in Fig 2A which terminate on node E. All paths can be overlayed together forming a rooted tree graph which is referred to as the input tree. In this example only the first three layers of the input trees of nodes E and F are shown, but it can already be seen that their input trees are identical if all the labels of the nodes and edges are ignored. Nodes with isomorphic input trees eventually synchronize regardless of synaptic delay or small variations there parameters [67, 68]. Colored lines indicate the morphism between trees. An input tree can be annotated by the number of edges between specific nodes at specified layers, the edge from B to F is shown for both input trees where it appears at different layers.

distinguishment of unique and not isomorphic input trees is $n-1$, where $n$ is the total number of nodes in a graph $G$ [35].

Nodes with the same input tree are in the same *fiber*, which are projections of a morphism called a *graph fibration* [28]. A graph fibration $\varphi$ between two graphs $G = (N_G, E_G)$ and $B = (N_B, E_B)$ is satisfied when for any $e_B \in E_B$ and any $v_G \in N_G$ with $\varphi(n) = h(e_B)$, there is a unique $e_G \in E_G$ such that $\varphi(e_G) = e_B$ and $h(e_G) = v_G$. A *fiber* is a set of nodes (cluster cell) with isomorphic input trees in $G$, mapped via $\varphi$ onto a node in $B$, where $G$ is the total space and $B$ is the base. This paper focuses on the *minimal graph fibration*, collapsing a graph into its *minimal base*, where the base contains one node for each unique input tree, resulting in a trivial partitioning of $B$ and a *minimal* balanced coloring of $G$ [28]. A partitioning with $k$ cluster cells that obey equitable partitioning is minimal if no other partitioning with fewer than $k$ clusters satisfies the equitable partitioning condition [31]. An example of this can be seen in Fig 3. Throughout this paper we refer to the *minimal graph fibration* as the *minimal balanced coloring* of a graph.

Based on theory nodes in the same fiber synchronize, meaning that the associated admissible ODEs have the same value at the same time [28, 36]. Also based on theory, there are no restrictions on the differences in behavior between two nodes in two different fibers, and it is possible for two or more fibers to synchronize [27], such is dictated by the set of equilibrium solutions of an ODE system. Connected nodes in the same fiber may have phase shifts under

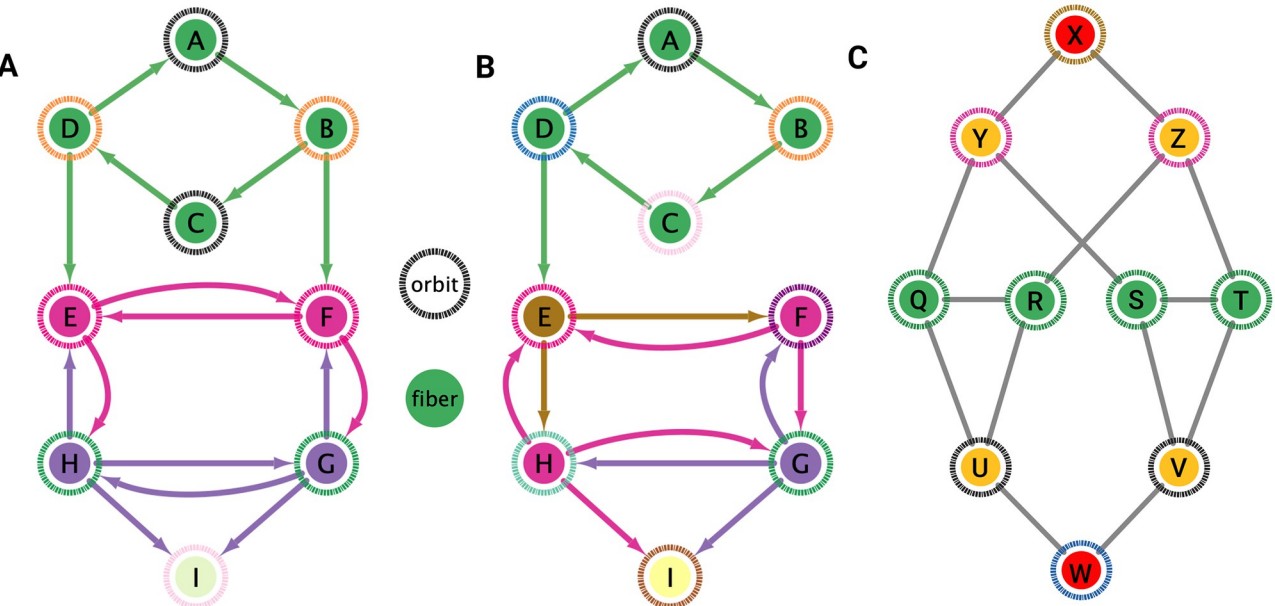

**Fig 2. Fiber and orbits.** The inner color of a node is a representation of the fiber to which said node belongs to and the color of the ring indicates to which orbit such a node belongs to. (A) Graph with 4 fibers and 5 orbits. The separation of fiber and orbits among the 4 upper nodes occurs due to the presence of outgoing edges into nodes E and F. (B) This graph only differs from A by the removal of the edge *B→F* exemplifying how restrictive orbit symmetries are. There are only trivial orbits but there still non-trivial fibers. (C) Although it is more common to have a matching of the partitionings produces by fibers and orbit symmetries in a given undirected networks there can exist situations in which these two are not matching as seen in this example. In this simple example there is a breakage between the fiber and orbit symmetries due to the in-existence of a permutation action which could swap nodes X, Y and Z respectively with their fiber symmetric counterparts W, U and V meanwhile preserving its structure and adjacency matrix.

the same frequency or an integer multiple of a base frequency, but only if the nodes have rotational symmetry, and the phase shift is a rational fraction of $2\pi$ [31]. Although the natural occurring biological variations from neuron to neuron can be expressed as variations in the same set of parameters among the admissible ODEs and the physical propagation of signals between neurons can be incorporated through different time retardation terms. All these modifications do not necessarily destroy the expected synchronization of nodes in the same fiber, but do indeed reduce the parameter space of the equilibrium solution where such is attainable [67, 68]. Therefore we forgo integrating these for the sake of simplicity of our models.

We find the fiber partitioning algorithmically by initially coloring all nodes and arrows with a unique color. After that, the algorithm recolors all nodes with the same number of colored inputs with a new color, including their outgoing arrows. These procedures continue until no recoloring is possible. This is the algorithm developed by Kamei & Cock [69], which is a generalization of the algorithms developed by Belykh and Hasler [70] and by Aldis [66] and implemented through the recreated code in [38].

**Automorphism and their orbit partitioning.** The second method relies on the set of automorphisms of a graph, which is a bijective function $\sigma$ from a graph onto itself preserving the structure of the graph, specifically its adjacency matrix $A$. The set of all automorphisms of graph $G$ along with the operation of composition forms the automorphism group $Aut(G)$ [71]. Specifically, an automorphism is a permutation of the vertices of the graph that preserves the adjacency relationships between them. If $G$ is a graph with vertices $\{v_1, v_2, \ldots, v_n\} \in N_G$ and with edge set $E_G$, then an automorphism of G is a permutation $\sigma$ of the node set such that if $(v_i, v_j)$ is an edge in $G$ so too is $(\sigma(v_i), \sigma(v_j))$ an edge in $G$. An automorphism function $\sigma$ can take

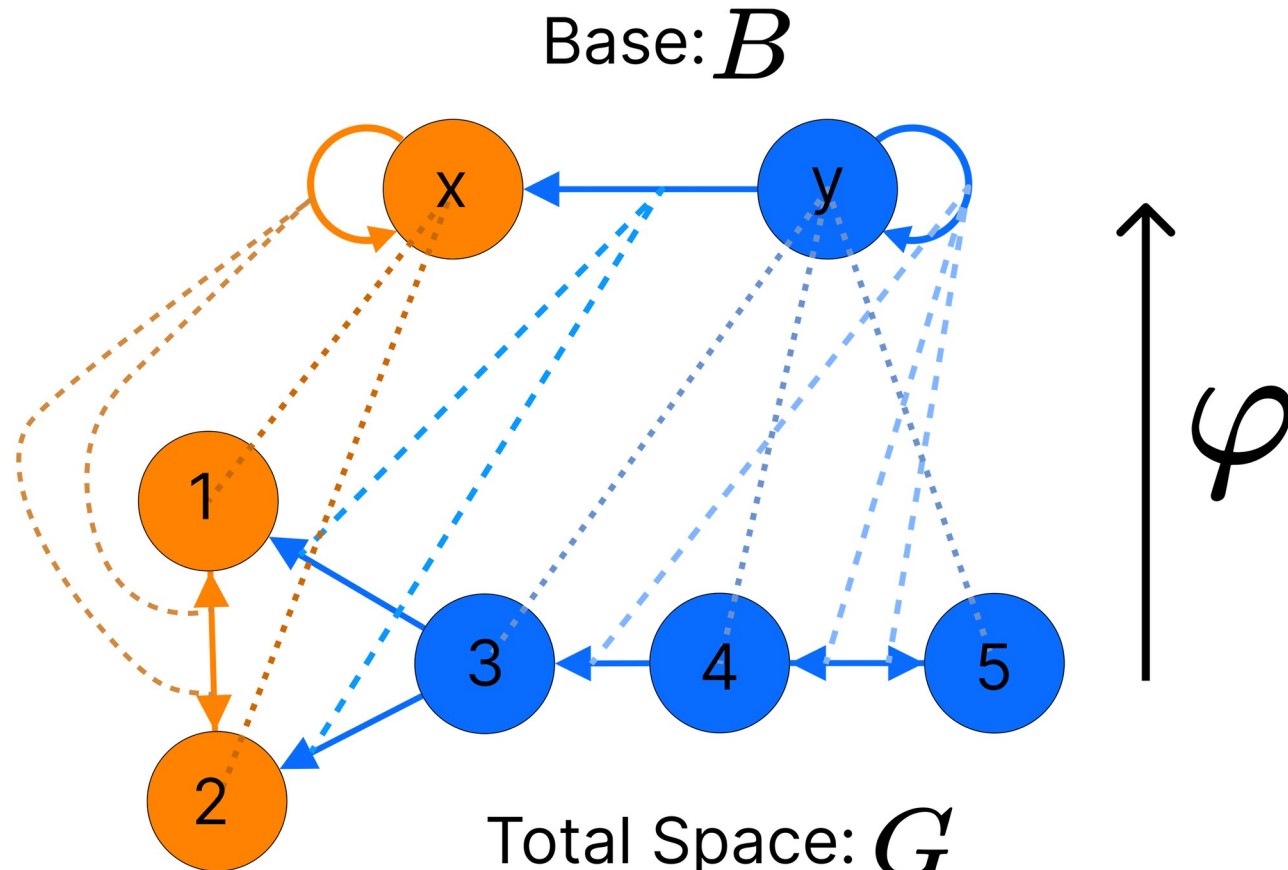

**Fig 3. Graph fibration $\varphi$: $G \to B$.** The network at the bottom ($G$) shows an example for a graph where the color of its nodes represents the fiber to which they belong. Notice that all nodes with the same (e.g. {3, 4, 5}) color are collapsed on to one representative node (y) in the network above called the base. This collapsing process is known as the lifting property of a graph fibration $\varphi$. This property also applies to edges, notice that the edges in the total space (e.g. $\{e_G^{3\to1}, e_G^{3\to4}\}$) which are lifted to one representative edge in the base which is situated between nodes of the same color as in the total space ($e_B^{y\to x}$).

the form of a symmetric permutation matrix $P$ where matrix is considered to be an automorphism of an adjacency matrix $A$ if the following holds:

$$PAP^{-1} = A, \tag{5}$$

where $P^{-1}$ is the inverse of $P$. Additionally it is invertible meaning that $P = P^{-1}$ or $PP^{-1} = 1$ allowing Eq (5) to be rewritten as:

$$[P, A] = AP - PA = 0. \tag{6}$$

The permutations of the automorphism group can be grouped into sets that form normal subgroups meaning that a permutation from one set can commute, as in Eq (6), with the permutation of another set (disjoint unions) [34]. In other words, applying one of these permutations to a graph it only permutes a closed set of nodes, leaving the rest of the graph's nodes intact.

If when $\sigma$ is applied on a graph's adjacency matrix $A$ as in Eq (7) and it doesn't change it, then multiple consecutive applications won't either.

$$\sigma(A) = PAP^{-1} \qquad (7)$$

Although an automorphism's consecutive application preserves a graph's adjacency matrix, it does shuffles node labels. So, a node's label always returns to its original after visiting other nodes. Given a graph $G$ and $\sigma$, the orbit of node $v$ under $\sigma$ is the set of all vertices that can be reached by applying $\sigma$ repeatedly. That is, the orbit of $v$ is the set nodes

$$\{\sigma^k(v) : k \in \mathbb{Z}\} \qquad (8)$$

where $\mathbb{Z}$ denotes the integers numbers. The orbit of node $v$ includes all vertices equivalent to $v$ under the action of $\sigma$, transformed by applying $\sigma$ $k$ times. The set of nodes belonging to an orbit are said to have the same *orbit coloring* [32, 72] were studies in dynamical systems show that network automorphisms lead to the synchronization nodes in an orbit [31, 69, 73, 74].

Orbit partitioning results from analyzing the complete set of automorphisms of a graph *Aut* (*G*). Orbits can be revealed not only by repetitively applying a single automorphism but also by applying the set of automorphisms *Aut*(*G*) consecutively in all possible orders and multiplicities (e.g., $P_3P_1P_2P_2$). This approach allows for a wider range of transformable nodes to be detected. The procedure of how this is explicitly done is out of the context of this paper, but the process of uncovering orbit coloring partitions is done using SageMath [75], an open-source mathematics software that utilizes various Python packages, including Nauty [76].

The orbit partitionings that emerge from this are equitable partitioning which are cluster cells, stressing that an orbit coloring is always an equitable partition, but the reserve is generally not true [31, 32]. This is because, by definition, orbit partitioning requires the conservation of both in-degree, requested for the equitable partition, and out-degree. As such, an equitable partition may not be an orbit partition since the out-degree condition may not be met.

**Fibers vs orbits.** As discussed above the equitable partitionings of fibers and orbits are related to the concept of symmetry. The orbits obtained from automorphisms are analogous to the fibers obtained from symmetry fibrations. Orbits, however, are a more restrictive form of graph symmetry as these must satisfy in-degree and out-degree edge constraints. In contrast, fibers must only satisfy in-degree constraints.

Fibration symmetries in a graph have fewer constraints than automorphisms, resulting in fewer partitionings in the graph. Orbits are a subset of fibration symmetries, which additionally conserve the out-degree structure of the adjacency matrix. This means that every orbit is also a fiber, and the in-degree constraint present in the orbit partitioning, which is the main constraint in fiber partitioning, leads the nodes in the same orbit (cluster cell) to become synchronous [33]. However, note that a fiber is not necessarily an orbit.

In Fig 2A, an example of a graph with three fibers and four orbits is shown. The existence of more fibration symmetries than automorphisms generally leads to having fewer (or equal) fibers than orbits. In other words, having more symmetries implies having fewer cluster cells. The green nodes have the same number of in-degree edges (belonging, therefore, to the same fiber) but some have different number of out-degree edges. Nodes *B* and *D* each have an out-degree equal to 2. Another way to conserve the structure of this graph other than applying the identity transformations is by rotating $A \rightarrow B \rightarrow C \rightarrow D \rightarrow A$ twice in this order, and in concurrence reflecting horizontally all the nodes below the green nodes. Without applying this reflection, node *B* would have one outgoing edge into node *E*, therefore not preserving the original graph's structure. Notice that any permutation for the upper four nodes would still preserve its fiber symmetry as these would continue to only receive one green in-degree edge.

As an additional example, the removal of edge $B \rightarrow F$ as seen in Fig 2B destroys the orbit symmetry of the graph. No rotation between the upper four nodes would allow node $D$ to have an outgoing edge to node $E$ other than its original position. There also does not exist any permutation of the lower five nodes that would keep the structure intact all due to the removal of $D \rightarrow E$.

## Building blocks

As an example, number theory shows that every natural number can be represented as a unique product of prime numbers [77]. Prime numbers are considered the fundamental building blocks of natural numbers. This concept is extended to group theory, where finite groups can be broken down into simple subgroups [78]. This abstract example has implications for natural systems due to the relationship between group theory and symmetry. A similar relationship can be applied to biological networks, as shown in [39, 79–81]. In this paper, besides partitioning network nodes by fiber (and orbit) symmetries, we also try to categorize all fibers into distinct and succinct fundamental units. These fundamental units (or otherwise named building blocks) relying on the (topological) properties of the input tree of a given fiber as previously discussed in [26]. Despite the diverse range of input tree topologies, they share common structural characteristics that allow us to categorize these into fiber building blocks (FBBs).

**Circuits.** In this paper we show the fiber building blocks of the chemical networks. In order to do so and explain it in a more detailed manner each of the networks will be broken into multiple sub-graphs. Where we refer to these sub-graphs as *circuits* and for a graph with $k$ fibers there will be $k$ circuits. These circuits are produced by collapsing any of these chemical network graphs (total space) into their base graph representation and retaining the unique fibers present in the input tree of a fiber under consideration. Take the Backward Chemical network (a.k.a. B-Chem) as seen as a graph in Fig 4A as an example. In this graph the color of the nodes represents the fiber to which they belong (more on this particular network's partitioning ahead). Neurons DA05, DA08, DA09, VA06 and VA11 are all the nodes contained in what we can call fiber Blue5 standing for the 5 blue nodes composing it. The input tree of fiber Blue5 contains nodes belonging to, using the previous nomenclature, fibers Orange4, Pink2, Cyan2, Mustard2 and Gray4. The circuit associated with fiber Blue5, call this the main fiber or the fiber under consideration, can be observed at the left of Fig 5. It is produced by collapsing the B-Chem network and retaining the fibers with their outgoing edges that compose the input tree of fiber B5, call these other fibers the induce fibers.

**Fiber building block synthesis.** A circuit is further broken down into its fundamental building block which here and as in [26] we refer to as fiber building blocks. A circuit has as many FBBs as is has induce fibers plus its main fiber, 6 FBBs for the B-Chem as it is seen in Fig 5. As mentioned for each fiber in a graph there exists a fiber building block (FBB) associated with it which is a sub-graph $S \subseteq G$ induced by the following nodes [26]:

- all the nodes in the fiber (nodes with the same minimal balanced coloring),

- immediate in-degree neighboring nodes of all the nodes in the fiber,

- nodes belonging to the shortest loop that include a node in the fiber which are not a self-loop,

- if all points above lead to two or more disconnected graphs, the nodes composing the shortest path connecting each pair of disconnected graphs are included. These types of FBBs are dubbed composite fiber building blocks which are further explained below.

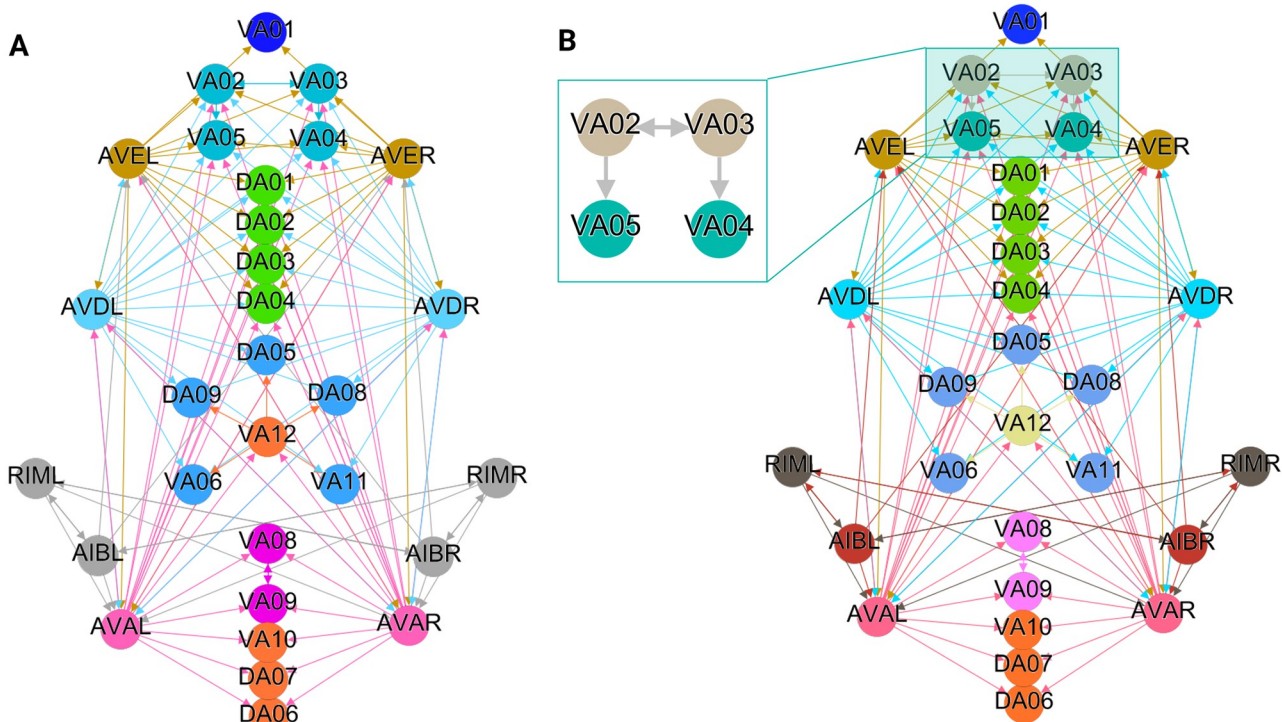

**Fig 4. Backward chemical synapses network partitioning results.** The partitioning of nodes into different synchronous groups is depicted by the color of the nodes. (A) The minimal balanced coloring results show 10 distinct colors/partitions for the 30 neurons involved. (B) The orbit coloring results for this network where the 30 neurons have been partitioned into 13 distinct partitions. The colored shaded areas show the normal subgroups that form the automorphism group of this graph ($Aut(\mathcal{G})$) with their respective symmetry group represented by the same colored symbol. (C) An example of neurons with different minimal balanced coloring and orbit coloring. The latter has a permutation symmetry $S2$ swapping VA02 with VA03 and VA05 with VA04 simultaneous which preserves the structure of this network.

For all nodes that are considered in the second point above we refer to these as *regulator* nodes. If the nodes in the main fiber are found to send information back into the fiber then these too are considered as regulatory nodes.

**Fiber numbers.** The FBBs that make up a circuit, are graphs in their own right and by such can also be represented by an input tree. These input trees are the bases of finding the building blocks of biological networks such as in the chemical synapses connectome of the *C. elegans*. Our approach is to categorize these FBBs through their structures and through such capture the fibrational composition of these networks. Below we explain how these can be decomposed into two features that numerically quantify the structure of the FBB's input tree. We first introduce and explain the feature followed by an example using the FBB of the cyan node found in Fig 5 (top sub-graph of Layer 2).

A feature of FBB that can be use to classify them is related to the number of nodes that are present at every layer of the FBB's input tree. Specifically how the number of nodes grow or remain the same relative to consecutive layers. This structure for a given input tree $\mathcal{T}(v)$ can be represented by a numerical sequence $a_i$ that denotes the number of nodes for every layer $i$. This sequence also corresponds to the number of walks terminating on the single node $v$ at the very top of the input tree, a.k.a. the root of the input tree. The exact number of nodes at every layer does not matter as much as how the number of nodes change per consecutive layer. This can be captured by the *branching ratio*, or *common ratio*, which can be found by taking the

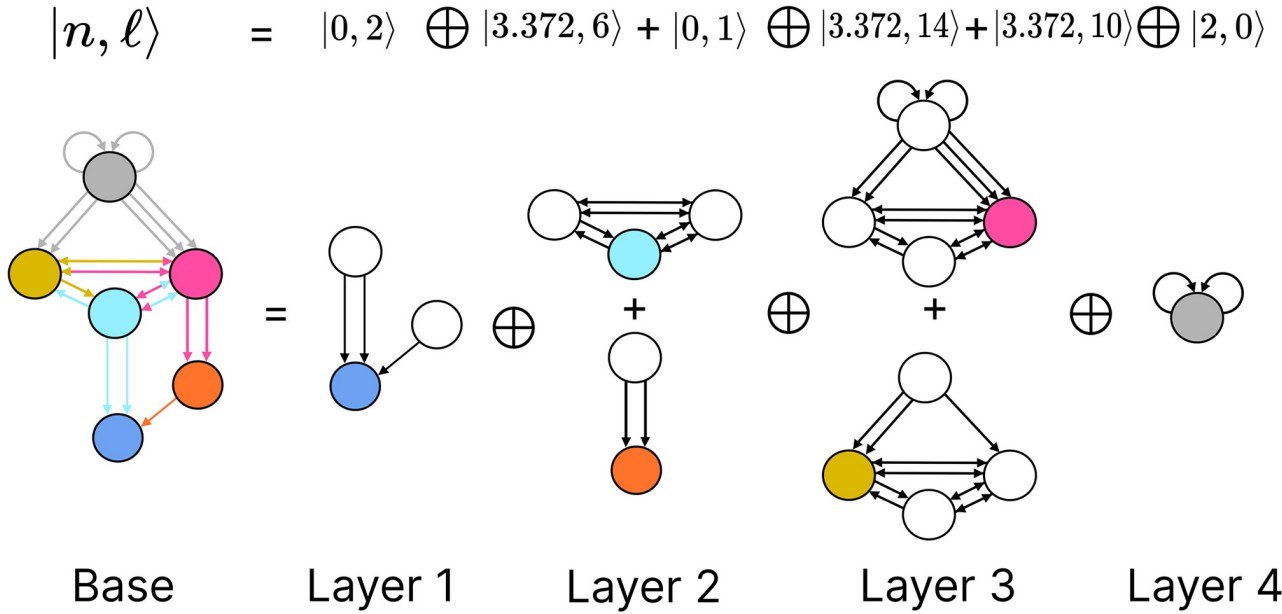

**Fig 5. Fiber building block decomposition.** Present here at the most left is the base of the circuit induced by the input tree for the fiber containing neurons DA05, DA08, DA09, VA09, and VA11 (blue node) present in the backward chemical network as seen in Fig 4A with the same color scheme. The first input into this fiber are provided from the orange (VA12) and cyan (AVD) nodes where these form the first layer of the multilayered block. The input tree of the network formed by these nodes give a fiber number of $|0, 2\rangle$ ($\ell = 2$ as multiplicity of edges are ignored). The next layer is composed by the elementary fiber building blocks of the input fibers in the previous layer. This notion is carried out at every layer until all fibers have been shown only once. Layer 2 already shows a complex elementary FBB for the cyan node with fiber numbers $|3.372, 6\rangle$. Layer 3 too contains elementary FBB with a similar fiber number as the cyan node in layer 2; the only difference here is that these two contain different number of inputs therefore the $\ell$ number is different in these two. Additionally in layer 3, self-loops of the gray node are present for the upper elementary FBB as the nodes forming this fiber are composed by neurons AIB and RIM which form a square network and which all send signals to the pink fiber. As for the gray fiber in the elementary FBB at the bottom it is only formed by neurons AIB which have no connections between them and therefore their collapse does not produce self-loops. The fiber building block induced by the input tree of the blue fiber concludes with a 4th layer which contains 4 neurons connect in a square. These have no external input which leads to it having fibers numbers $|2, 0\rangle$ which captures the duplication of nodes at each layer in their input tree.

limit of $a_{i+1}/a_i$ as $i \to \infty$. This ratio is denoted by $|n\rangle$ which captures the number of loops in the input tree of a fiber. The symbol $|n\rangle$ is called a *fiber number* and can be an integer or an irrational number, with the latter indicating complex branching due to nested loops which are referred to as "Fibonacci fibers".

Using our example, the first layer has only one node as is the case for any input tree, in this case the cyan node. The second layer is composed by 3 nodes (as the cyan node has 3 in-degree edges). We obtain the number of nodes in the third layer by calculating the total number of in-degree edges for all the nodes present in layer 2. For this case one of the nodes in layer 2 receives 3 in-degree edges, while the remaining two nodes (repeats) receive 4 in-degree edges making for a total of 11 nodes. Using the third and second layer we obtain a branching ratio of $a_3/a_2 = 11/3 = 3.\bar{6}$, calculating this branching ratio for layers evermore faraway from this input trees' root it reaches a limit value of $3.372\ldots$.

Another property that is used to characterize a FBB's input tree $\mathcal{T}(v)$ for a node $v$ is the number of unique trails with no repeating edges which terminate on node $v$. Where a trail is a walk in which all directed edges are distinct (transversed once). The number of trails is calculated in the base graph induced by the nodes that compose the input tree disregarding edge multiplicity between nodes. This property is symbolized by $|\ell\rangle$ which is another fiber number and together with $|n\rangle$ are enough to categorize the topology of input trees where the two can be agglomerated into $|n, \ell\rangle$ which are called *fiber numbers*.

Using the cyan FBB in Fig 5 we explain how to calculate the numerical fiber number $|\ell\rangle$. Before anything lets call the node above and to the left of cyan node the $M$ node (for the color mustard), and the node above and to the right the $P$ node (for the color pink). All the unique trails terminating in the $C$ node are: $\{M \to C; P \to C; M \to P \to C; P \to M \to C; P \to M \to P \to C; M \to P \to M \to C\}$. The previous fiber number $|n\rangle$ is ultimately another quantification of the trails present in a FBB's input tree, particularly of closed trails that result in loops, where loops of different length can be "nested" in each other resulting in Fibonacci fibers. Using our cyan fiber building block it can be seen that it has loops of length 2, 3, and 4 of which each is nested in the other. Where an example of a loop of size two is $P \to M \to P$, a loop of size 3 can be the closed walk $C \to M \to P \to C$ and an example of a loop of size 4 is $C \to M \to P \to M \to C$.

**Multilayering.** Simpler fiber building blocks can be combined to create multilayered composite blocks needed when the topology of a circuit or FBB cannot be fully captured by one fiber number pair $|n, \ell\rangle$. In such cases, the $|n, \ell\rangle$ of the input tree of each regulator of an FBB is added to the $|n, \ell\rangle$ of the input tree of the main FBB itself, creating an addition between two adjacent layers. This addition is performed in a left-to-right ordered sequential manner, resulting in the following notation:

$$|n_1, \ell_1\rangle \oplus |n_2, \ell_2\rangle + |n_3, \ell_3\rangle \oplus \dots. \tag{9}$$

The symbol $\oplus$ represents an addition of FBBs between adjacent layers and the symbol $+$ is used to indicate that the two FBBs next to this symbol are in the same layer. In the example above the fiber number $|n_1, \ell_1\rangle$ represents the FBB at the first layer. Fig 5 is an example of multilayered FBB composed of regular and Fibonacci fiber building blocks. Overall these types of combination of fiber building blocks aid in determining the order of complexity among fibers.

## Admissible ordinary differential equations

We compare results of fiber and orbits with cluster synchronization obtained from simulating interacting neurons. Interactive nodes are simulated by mapping admissible in-degree dependent coupled ordinary differential equations (ODE) of the form in Eq (10) to each node in our networks [28, 30, 31, 36, 74]. The $i$-th neuron's input is the $i$-th column of an adjacency matrix $\tilde{A}$, where the $j$-th row in the column represents the strength of the input from neuron $j$ to neuron $i$, and this strength is the weight of the edge from neuron $j$ into neuron $i$.

$$\frac{dV_i}{dt} \equiv \dot{V}_i = \mathbf{f}(V_i) + \tilde{\sigma} \sum_{j=1}^{n} \tilde{A}_{ji} \, \mathbf{g}(V_i, V_j) + \sigma^{ext} I_i^{ext}, \quad i = 1, \dots n \tag{10}$$

Each $i$-th neuron of a network has a voltage $V_i$ associated with it, representing the membrane potential of the $i$-th neuron. The rate of change of this voltage $V_i$ is given by an ODE of the type in Eq (10). As the equation shows, $V_i$ is affected by the voltage of other neurons coupled with it with a strength associated with the sum on the right side of Eq (10).

To emulate the activation of neurons involved in the forward or backward locomotion of *C. elegans*, our models provide external stimuli through the term $I_i^{ext}$ to only some neurons in each network. This action is called driving or stimulating the network. The left and right versions of an inter-neuron pair, which govern the activation of the downstream motor-neurons DA, DB, VA, and VB responsible for activating wall muscles that produce locomotion, are mainly selected to receive the external stimuli through $I_i^{ext}$ [52]. All nodes without external input stimuli have their $I_i^{ext}$ set to zero. The time constants represented by the symbol $\sigma$ control the rate of change of the ODE components. Neurons that are considered to be synchronous

through their graph analysis (like inter-neuron pairs) must receive the same input or else these will not attain synchronicity.

The smooth internal state function $\mathbf{f}(\cdot) : \mathbb{R} \to \mathbb{R}$ and pairwise interaction function $\mathbf{g}(\cdot, \cdot) :$ $\mathbb{R} \times \mathbb{R} \to \mathbb{R}$ can be either linear or nonlinear, and are present for both types of interactions in the *C. elegans*: chemical synapses and gap junction interactions. The gap junction interaction function is linear, while the chemical synapse interaction function is nonlinear. The internal state function $\mathbf{f}(\cdot)$ brings a neuron back to its resting state voltage $V_{rest}$. For the chemical synapse interactions, two types of functions are explored: the Chem type I model based on [82, 83], and the Chem type II model taken from [84–86]. These lead to three types of ODE interactions applied to their corresponding networks:

$$\dot{V}_i = \mathbf{f}(V_i) - \alpha^{\text{gap}} \sum_{j=1}^{n} \tilde{A}_{ji}^{\text{gap}} \left( V_i - V_j \right) + \alpha^{\text{ext}} I_i^{\text{ext}} \quad \textit{Gap junction} \tag{11}$$

$$\dot{V}_i = \mathbf{f}(V_i) - \alpha^{\text{chem}} \sum_{j=1}^{n} \tilde{A}_{ji}^{\text{chem}} \Phi(V_j) \left( V_i - V_{s,j} \right) + \alpha^{\text{ext}} I_i^{\text{ext}} \quad \textit{Chem type I} \tag{12}$$

$$\dot{V}_i = \mathbf{f}(V_i) - \alpha^{\text{chem}} \sum_{j=1}^{n} \tilde{A}_{ji}^{\text{chem}} s_j \left( V_i - V_{s,j} \right) + \alpha^{\text{ext}} I_i^{\text{ext}} \quad \textit{Chem type II} \tag{13}$$

with support functions:

$$\mathbf{f}(V_i) = -\alpha^{\text{leak}} (V_i - V_{\text{rest}}) \tag{14}$$

$$\frac{ds_i}{dt} \equiv \dot{s}_i = a_r \Phi(V_i) \left( 1 - s_i \right) - a_d s_i \tag{15}$$

$$\Phi(V_i) = \frac{1}{1 + \exp(-\gamma(V_i - V_i^{\text{threshold}}))} \tag{16}$$

$$I_i^{\text{ext}} = I_{\text{drive}} + I_{\text{osc}} \sin 2\pi ft + I_{\text{noise}} \tag{17}$$

The term $I^{ext}$ in Eq (17) consists of three parts: $I_{drive}$ is a positive constant current; $I_{osc}$ is an oscillatory term and a Gaussian random walk term $I_{noise}$ (with standard deviation of 1 scaled between $0.1pA$ to $-0.1pA$). The oscillatory stimuli frequency of the driven neurons was selected between 1Hz to 3Hz to approximate the swimming frequency of an adult wild-type *C. elegans* in agar [87]. The external stimuli function $I^{ext}$ was enforced to be the same for neurons with the same balanced coloring as to preserve their input symmetry.

A consensus agreed upon depiction of dynamics of neurons would have independent noise for each neuron. If this case were applied, neurons belonging to the same fiber will indeed no longer have an exact synchronization $V_i(t) = V_j(t)$. Depending if the strength (amplitude) of the noise term is kept small relative to the constant driving stimuli or the epsilon term in the LoS measure (measure introduced in the following section), the latter will give a result close to 1 indicating near synchronicity. As it will be seen, in our models inter-neuron pairs are the ones to receive external stimuli and therefore drive the network. If each inter-neuron receives independent noise, the synchronicity of all other neurons will not be affected by this implementation. This is due to the fact that all of the networks are left-right symmetric giving way to neurons in each fiber receiving both signals provenient from the driven inter-neuron pair. By

such there are no neurons that only receive a signal from only one of the driven inter-neurons which would lead to a breaking of the expected fiber partitionings.

Reiterating the models used here do not have time delays to represent the finite time of signal propagation between neurons, it has been shown that neuron ODE models can still synchronize when this effect is taken into consideration [67]. Additionally the parameters used and explained below need not be exactly the same between neurons for synchronization to occur, although it can reduce the system's capacity to achieve synchronization [68].

**Ordinary differential equation parameters and implementation methods.** The parameters of these ODEs shown above are taken from Kunert *et al.* [85]. The parameter $\alpha^{leak}$ is a time constant with a value of $10s^{-1}$ and is associated with the rate of decay of the neuron's membrane potential. This is derived from the leakage conductance per surface area $g_L$ times the average surface area of an inter-neuron $S$ (based on Varshney [8]$g_L \cdot S$ is equal to $0.01nS$) divided by the membrane capacitance of a neuron (Varsheny *et al.* [8] defines $C$ to be $1pF$). The time constants for chemical synapses and gap junctions have the same value of $g_L \cdot S/C = 0.1nS/1pF = 100s^{-1}$ where these fall within the value range from Koch's book on biophysics [88].

The constant $\gamma$ is associated with the steepness of the sigmoid function which is set to $2ln(0.1/0.9)/36mV$ in accordance to Wicks *et. al* [53], which defines this constant to be representative of a change of the value of the sigmoid function between 0.1 to 0.9 within a range of $36mV$. This value is close to that reported by Kunert [85] so it is rounded to $125V^{-1}$ to match Kunert's value.

The synaptic activity variable depicts the activity magnitude at a synaptic junction, and its behavior is affected by the neuron's membrane potential as well as the parameters determining the increase and decrease of synaptic activity. The terms $a_r = 1s^{-1}$ and $a_d = 5s^{-1}$ associated with the Chem type II model correspond to the synaptic variability's rise and decay times [85]. The equilibrium value for the synaptic variability term comes from setting Eq (15) to zero and having the sigmoid equal to 0.5; this leads to a value of $s_{eq} = ar/(ar + 2^*ad) = 0.09$. Finally $\alpha^{ext}$ is simply set to $1/C$ and $V_{s,j}$ takes two values: $0mV$ for excitatory vs $-70mV$ for inhibitory presynaptic neurons [89]. For this study, we considered all chemical synapse connections to be excitatory (although our code accepts networks with inhibitory connections). The units of $I^{ext}$ are $pA$ to match the units ($V/s$). All neurons in the simulation conducted in this paper have the same parameters as the different families of motor-neurons are known to be the most similar among all the differently distinct electrophysiological group of neurons [49]. We extend this notion to the parameters of the inter-neurons to have a simpler system to work with, as well due to the nature of our sub-networks being mainly composed by motor-neurons.

We set the threshold voltage $V^{threshold}$ for each neuron to the corresponding solution obtained by setting Eqs (12), (13) and (15) to zero and Eq (16) to 0.5; where the solution is the stable point solution for this ODE model. This procedure (Gaussian elimination) is similar to that in Wick *et al.* [53] where in such $I^{ext}_i$ is set to 0 where as we set it to $I_{drive}$ (the mean value of an undulatory or of a noise stimuli is 0, therefore, these are not included). This recipe finds the solution of the threshold voltage to be the one below

$$V^{\text{threshold}} = A^{-1}b \tag{18}$$

such that $A$ is given by

$$A_{ii} = 1 + \frac{1}{\sigma^{\text{leak}}} \sum_{j=1}^{n} (\sigma^{\text{gap}} \tilde{A}_{ji}^{\text{gap}} + c\sigma^{\text{chem}} \tilde{A}_{ji}^{\text{chem}})$$

$$A_{ji} = -\frac{1}{\sigma^{\text{leak}}} \sum_{j=1}^{n} \sigma^{\text{gap}} \tilde{A}_{ji}^{\text{gap}} \tag{19}$$

and $b$ by

$$b_i = V_{\text{rest}} + \frac{1}{\sigma^{\text{leak}}} \left( \sum_{j=1}^{n} c\sigma^{\text{chem}} \tilde{A}_{ji}^{\text{chem}} V_{s,j} + \sigma^{\text{ext}} I_i^{\text{ext}} \right). \tag{20}$$

where the term $c$ is equal to $s_{eq}$ for the Chem type II model or 0.5 for the Chem type I model. It is important to point out that threshold voltage is respectful of the in-degree nature of Eq (10). Further in the paper $V^{threshold}$ is used in conjunction with the Jacobian to determine the stability of the solutions.

The ODE dynamics were evolved and computed using the Runge-Kutta-4th method with a time step $dt$ of 0.1 milliseconds. Noise dynamics were implemented through a modified stochastic Runge-Kutta method as proposed in [90], where the update time step for the noise is $dt^{1/2}$. A link to a user-friendly Matlab app developed in-house can be found in the Data Availability section. This app can reproduce any of the dynamics presented in this paper with the appropriate inputs. It can measure the level of synchronization (explained in the next section) and has additional tools for further experimentation.

## Synchronicity measure

Synchronicity can be quantified in several ways, including Pearson correlation, covariance, and cross-correlation. However, we propose a stricter metric based on the potential of two neurons $i$ and $j$ being fully synchronous if they have the same value within a certain time window as defined in Eq (21).

$$V_i(t) = V_j(t) : \forall t \tag{21}$$

To measure near-full synchronicity and distinguish between distinct potentials, we introduce the Level of Synchronicity (LoS), which utilizes a time-averaged Gaussian kernel distance [91]. The LoS metric enables us to differentiate between situations of nearly equal potentials and those with no synchronicity at all.

$$LoS_{ij} = \frac{1}{T} \sum_{t=time\ step}^{T} \exp\left\{ -\frac{[V_i(t) - V_j(t)]^2}{2\sigma^2} \right\}. \tag{22}$$

In the equation above, $T$ is the total amount of time steps on which $LoS$ is applied between the signals of neurons $i$ and $j$. The range of the $LoS$ function is [0, 1] where a value of 1 indicates full synchronicity, whereas a value of 0 would indicate no synchronization. The parametric term $\sigma$ serves as a scale to define a benchmark for closeness between two points.

To apply the $LoS$ metric to our simulations, we allow each network to reach a stable state after the initial transients by running it for a sufficient amount of time. Attaining such is guaranteed due to the equilibrium solutions of our ODEs being attractors as will be seen later in the paper. (Note: we use the term stable state equivalently to that of an equilibrium solution of an ODE system throughout this paper). We use the last second of simulation time to determine the synchronicity level, analyzing all pairs $(i, j)$ and producing an $LoS$ matrix for each network simulation. We use a value of $\sigma = 0.1mV$ for all simulations, such that a potential difference of $0.1mV$ between two neurons over a time window $T$ would yield an $LoS$ value of approximately 0.61. We chose $\sigma$ by starting at $10mV$, which produced a fully synchronous $LoS$ matrix for all networks and ODE models with no external stimuli. We then reduced $\sigma$ until the average of all $LoS$ pairs between two fibers in the network with the most fibers was below 0.001.

Besides the *LoS* measure, we used the Phase Locking Value (*PLV*) [92] measure at our last set of simulations to measure the amount of phase synchronicity. *PLV* quantifies the degree of synchronization or coupling between two undulatory signals. It is based on the idea of the difference between the instantaneous phase of two signals at each time point, and then computing the average over a certain time window. These values fall between 0 and 1, where 1 indicates two signals are in constant phase synchronicity (the two signals hold the same phase difference through a recording). A value of 0 indicates that the two signals have varying phases such that the difference between these is random and does not remain constant.

## Results

### Network partitions in the locomotion connectome

We focus on the forward gap junction (F-Gap), backward gap junction (B-Gap), forward chemical synapses (F-Chem), and backward chemical synapses (B-Chem) taken from [37] where each one of these are binary edges networks. In addition we also work with the integer edge weighted version of the aforementioned networks that respect the symmetrization done in [37] and their fiber partitionings. For the latter the weight of an edge between two fibers was taken to be the rounded-up integer of the average of all the edges present in the original connectome of [8] between the cluster cell of the fibers under consideration.

The resulting partitionings and their cluster cells are the same for both binary and integer weighted networks since the graph structure $G$ is the same regardless of edge type. However, the input tree for a fiber associated with a specific neuron is likely different between binary and weighted versions.

Additionally the adjacency matrix for each of the integer edge weighted networks is preserved under the permutation actions of the normal subgroups of the $Aut(\mathcal{G})$ of the respective binary network [37]. Notice that, in principle, automorphisms of weighted networks are different from automorphisms of binary networks: one weighted edge can easily break the symmetry of a binary network. As such, in a weighted network, the permutation needs to conserve adjacency taking into considering their weights. Due to the way the integer edge-weighted network has been constructed the orbits of the binary and integer edge-weighted networks are the same, meaning that the repetitive application of the same permutation on both the binary and integer versions of a network will result in a group of neurons visiting the same nodes of the network.

Figs 4, 6 and 7 show the results of the partitioning for the four different graphs under analysis. Fiber and orbit partitionings are the same for the forward and backward cases of the undirected gap junction networks (Figs 6B and 7 respectively). This is a direct consequence of the bidirectionality nature of undirected graphs where two nodes with the same fiber symmetry also preserve out-degree edges. Indeed, in an undirected graph, the in-degree (conditions for fibers and orbits) and the out-degree (condition for orbits) of a node has the same value. That is, every undirected link can be seen as an in and out-directed link, therefore, preserving the in-degree connectivity generally implies also preserving the out-degree, and therefore, the fibers can be expected to be the same as the orbits.

We emphasize that this is not always true. We can think of many cases of undirected networks where fibers are not the same as orbits, see Fig 2C. In this simple example, there is a breakage between the fiber and orbit symmetries due to the in-existence of a permutation action that could swap nodes X, Y, and Z, respectively, with their fiber symmetric counterparts W, U and V meanwhile preserving their structure and adjacency matrix. However, it is interesting that we find that in the gap junctions, the orbit partitionings equal the fiber partitionings, and fibrations symmetries do not add any more to what we can find with the

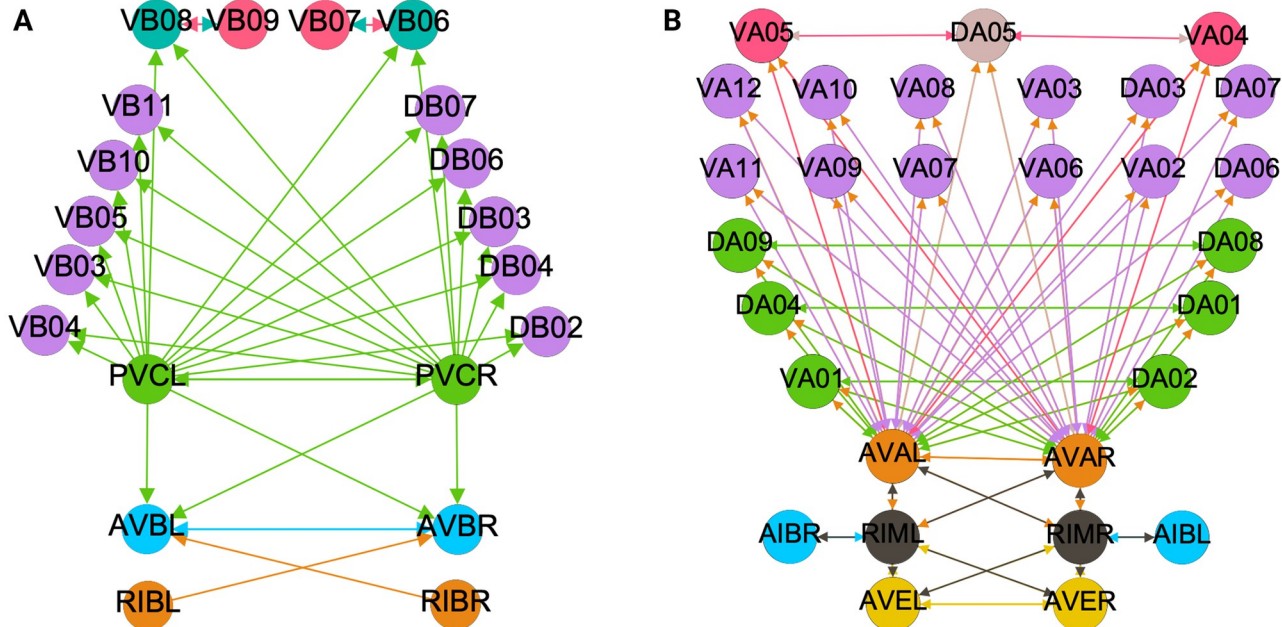

**Fig 6. Partitioning results for the forward chemical and backward gap networks.** Results for the minimal balanced coloring and the orbit coloring are same for these 2 networks. The colored shaded areas show the normal subgroups that form the automorphism group of these graphs ($Aut(\mathcal{G})$) with their respective symmetry groups represented by the same colored symbol. (A) The F-Chem network has its 22 neurons partitioned into 4 distinct colors. (B) The B-Gap network with its 29 neurons partitioned into 6 distinct synchronous groups.

automorphisms of these networks. Thus the structure of the gap is enough to be characterized by automorphisms [37] completely characterize these networks. In [37], we stop the analysis at the level of automorphisms, whereas here, we also find the associated orbits and their fiber building blocks.

The situation is different when the graph is directed, and as such, the in-degree of a node may differ from its out-degree. In these circumstances, the less stringent constraints of Fibrations result in a higher number of symmetries and, therefore, a smaller number of fibers. Obit and Fiber partitionings can also be the same in (directed) networks. These are equivalent when the automorphism group of the graph acts transitively on the set of fibers. Meaning that all the nodes of a fiber (for all fibers) should be able to be permuted with one another under the action of a permutation symmetry in the automorphism group of the graph [93].

In the F-Gap network (Fig 7), nodes VB03 and VB07 are an example where they have bidirectional connections with both AVBL and AVBR inter-neurons and DB01 and VB02 motor-neurons. They belong to the same fiber and orbit. If VB07 did not have out-going connections to the inter-neurons (AVBL and AVBR), and if VB03 did not have out-going connections to the motor-neurons (DB01 and VB02), then the fiber partitioning would remain the same. However, the orbit coloring would partition neurons VB03 and VB07 into their own unique colors since no permutation between these two nodes would preserve the connectivity matrix of the network. The minimal balanced coloring and orbit coloring are also the same for the F-Chem network, as the minimal balanced coloring for this particular network is modular and out-degree conserving in each of its partitions.

In contrast to other networks, B-Chem (Fig 4) exhibits differences between minimal fiber and orbit coloring. Consider the example of neurons VA02, VA03, VA04, and VA05 shown in Fig 4C, where each node receives the same number of inputs from inter-neuron pairs AVE,

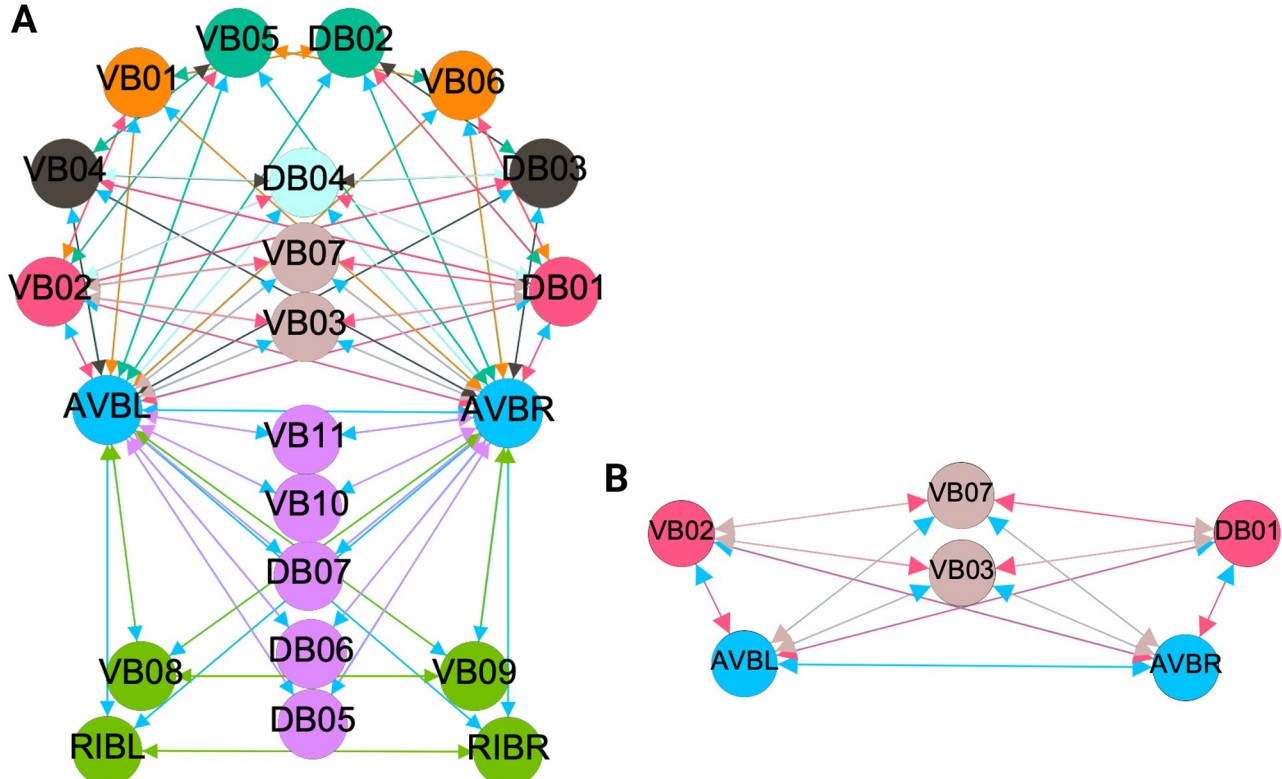

**Fig 7. Partitioning results for the forward gap network.** Results for the minimal balanced coloring and the orbit coloring are same for this network. The colored shaded areas show the normal subgroups that form the automorphism group of these graphs ($Aut(\mathcal{G})$) with their respective symmetry groups represented by the same colored symbol. (A) The F-Gap network with its 20 neurons partitioned into 6 distinct equitable partitions with one partition containing only one neuron. (B) Sub-graph used as a visual aid for the example found in the Network partitions in the locomotion connectome section.

AVD, and AVA. Any permutation among these nodes preserves the number of colored inputs they receive, resulting in the same fiber (minimal balanced coloring). However, node VA04 has no outputs, so permuting VA03 with VA04 would change the adjacency matrix of the sub-graph. The only way to preserve the out-degree of these four nodes under permutation is to simultaneously permute VA02 with VA03 and VA05 with VA04 ($\{\{VA02, VA03\}, \{VA04, VA05\}\}$), which is an instance of an orbit coloring.

Another example of differences between the minimal fiber partitioning and orbit coloring partitioning can be observed for VA12 and between L-R inter-neurons RIM and AIB. Besides the latter example, in all four networks, left-right inter-neurons pairs belong to their own unique partitioning.

## Fiber building blocks

The FBBs of the F-Chem are categorized by integer $n$ and low $\ell$ values, and are driven solely by neurons PVCL and PVCR. The first row of Fig 8 shows the only multilayered composite building block with fiber numbers $|1, 1\rangle \oplus |1, 2\rangle$, which represents VB07, VB09 and VB06, VB08, respectively.

The backward chemical network is composed of Fibonacci FBBs, formed by third layer inter-neurons AVA, AVD, and AVE, which receive information from first and second layered

**Fig 8. Forward chemical fiber building blocks.** Left column illustrates all the building blocks that form the binary F-Chem network. These sub networks arise as a consequence of its neurons belonging to a unique strongly connected component and all neurons with outgoing edges to it. The column named Fiber Base shows the collapsed version of the building block where numbers indicated the number of arrows between two nodes. The input tree here is a visual capturing of all the paths in the network in which the final node is part of the fiber under focus. The right most column features the "fiber numbers" $|n, \ell\rangle$ associated with each particular fiber. $n$ indicates the number of unique infinite paths and $\ell$ captures the number of "regulators" which are the neurons that only have out-going edges which form part of the input tree.

inter-neurons AIB and RIM [94] to form a loop circuit, as depicted in Fig 9. The branching ratio of Fibonacci FBBs in the backward chemical network is 3.3723. . ., which was not observed in a previous study of the E. coli's genetic network under this analysis [26]. This suggests a higher complexity in the neuronal circuitry of *C. elegans* compared to the genetic network of E. coli. Regardless, a FBB with an irrational number for $|n\rangle$ indicates that the block is composed of nested loops. In the case of Fibonacci building blocks, these have nested loops of length 2, 3, 4, and up to 5, as shown in Fig 5 at layers 2 and 3 where these form the elementary FBBs of the circuit for neurons DA05, DA08, DA09, VA06, and VA11 (same fiber).

The nested loop FBBs composed of AVA, AVD, and AVE can be compressed into smaller and simpler representational graphs using the Fibonacci base. For instance, the FBB associated with $|3.372, 6\rangle$ at the second layer of the network found in Fig 5 can be lifted through a graph fibration to the graph at the 8th row from the top of Fig 10. In this case, neurons AVD (cyan) and AVE (bronze) are collapsed onto one node that produces a self-loop as seen in the Fibonacci base, where the common ratio of the sequence $a_n$ produces a branching ratio of 3.372 for all nodes composing this network. Any FBB with a Fibonacci branching ratio can be compressed into a simpler graph, and therefore have a simpler sequence associated with it.

## Network simulations

In this section, we perform 3 types of numerical simulations according to the specified ODEs in the Admissible ordinary differential equations section and compare their outcomes to those predicted by orbits and fibers partitionings. Throughout these simulations we aim to showcase how the networks behave under different stimuli conditions and how different fibers synchronize with each other depending on this by contrasting the outcomes of simulation test 1 and 2. Within test 2 we seek to also determine the ranges in which the partitioning predictions are valid as external stimuli, if large enough, can induce instabilities. After such we explain how we stimulate these networks while remaining within the regime to not induce these instabilities. Simulation test 3 has the intention to inspect what happens to the expected fiber synchronization if the structure of the network (with external stimuli) remains the same but its edges take on evermore random values, we find that some version of synchronicity can still be captured.

**Simulation test 1 setup.**    We set the initial voltages to be normally distributed around the resting state potential of $-37mV$ and without external stimuli, which can be representative of a *C. elegans* during a state named lethargus. During this state, neurons such as ALM have shown low spontaneous activity and remain near their resting state [95]. The initial voltages are normally distributed around the resting state, with the standard deviation varied from 0 to 0.1. We applied this procedure to both types of chemical synapse models. We do not show results for the gap junction networks without external stimuli as they become globally synchronous, with nodes having a voltage equal to the resting state, due to nodes being able to be reached by any other node in the graph. For the Chem type II model, the standard deviation of the synaptic variable varied from 0 to 0.1 with a mean of $s_{eq}$ for every distribution of initial voltages. This setting allowed us to study the effects of the initial synaptic variables on the outcomes of synchronicity and inspect its stability.

**Simulation test 2 setup.**    For the second simulation test, we look into the synchronizations that arise in the networks under external stimuli as modeled by the ODEs found in the Admissible ordinary differential equations section. Contrary to the previous case, this setting allows us to explore if these symmetrical neural models of the *C. elegans* with gap junction connections synchronize to the predicted partitionings found via fiber partitionings, and if the B-Chem network dynamics separate into orbit or fiber partitionings when in an active state.

| Circuit | FBB Base | Input tree | Neurons | $|\varphi_d, \ell\rangle$ |
|---|---|---|---|---|
| | | | AIBL AIBR RIML RIMR | $|2,0\rangle$ |
| | | | AVDL AVDR | $|2,0\rangle \oplus |3.372,14\rangle + |3.372,10\rangle \oplus |3.372,6\rangle$ |
| | | | VA01 | $|2,0\rangle + |3.372,14\rangle + |3.372,6\rangle \oplus |3.372,10\rangle \oplus |0,1\rangle$ |
| | | | VA02 VA03 VA04 VA05 | $|2,0\rangle \oplus |3.372,14\rangle + |3.372,6\rangle + |3.372,10\rangle \oplus |1,3\rangle$ |
| | | | DA01 DA02 DA03 DA04 | $|2,0\rangle \oplus |3.372,14\rangle + |3.372,6\rangle + |3.372,10\rangle \oplus |0,3\rangle$ |
| | | | DA05 DA08 DA09 VA09 VA11 | $|2,0\rangle \oplus |3.372,14\rangle + |3.372,10\rangle \oplus |3.372,6\rangle + |0,1\rangle \oplus |0,2\rangle$ |

**Fig 9. Backward chemical fiber building blocks.** Left column illustrates all the building blocks that form the binary B-Chem network. These sub networks arise as a consequence of its neurons belonging to a unique strongly connected component plus all neurons which have out-going edges to such neurons in the strongly connected component. Starting from the third row many of the networks nodes are contained in a box where some of these have out-going edges indicated by the edges outside of the box. The column named Fiber Base shows the collapsed version of the building block where numbers indicated the number of arrows between two nodes (not fully captured in the first column). The input tree here is a visual capturing of all the paths in the network in which the final node is part of the fiber under focus. The last 3 input trees are omitted as these are too big to be figured here. The right most column features the "fiber numbers" $|n, \ell\rangle$ associated with each particular fiber. $n$ indicates the number of unique infinite paths and $\ell$ captures the number of "regulators" which are the neurons that only have out-going edges which form part of the input tree. Many of the fibers in this network are composite meaning that they are built from the "stacking" of multiple fibers.

| Fibonacci base | Branching ratio | Sequence |
|---|---|---|
|  | 1.0 | $a_n = a_{n-1}$ |
|  | $1 \quad if \quad n \quad is \quad odd$ <br> $2 \quad if \quad n \quad is \quad even$ | $a_n = 2 * a_{n-2}$ |
|  | 1.618... | $a_n = a_{n-1} + a_{n-2}$ |
|  | 2.0 | $a_n = a_{n-1} + 2 * a_{n-2}$ |
|  | 2.302... | $a_n = a_{n-1} + 3 * a_{n-2}$ |
|  | 2.561... | $a_n = a_{n-1} + 4 * a_{n-2}$ |
|  | 3.0 | $a_n = a_{n-1} + 6 * a_{n-2}$ |
|  | 3.372... | $a_n = a_{n-1} + 8 * a_{n-2}$ |
|  | 3.541... | $a_n = a_{n-1} + 9 * a_{n-2}$ |
|  | 4.0 | $a_n = a_{n-1} + 12 * a_{n-2}$ |

**Fig 10. Fibonacci bases.** The graphs presented here are the smallest possible, consisting of only two nodes, capable of producing an irrational common ratios. Additionally each of the examples above have a Fibonacci recursive sequence formula associated with them; reason for calling them Fibonacci bases. If any of the two nodes in these networks receive additional input from external nodes the recursive formula along with its branching ratio are not affected, although their sequences will differ. Due to this larger networks can be collapsed into one of these and still retain the same common ratio. As an example the elementary building block of the cyan node in Fig 5 has the same branching ratio of 3.372... as the Fibonacci base on the $8^{th}$ row. This elementary building block can be collapsed via a graph fibration into the Fibonacci base previously mentioned; ultimately yielding that the two networks are composed of the same fibers with isomorphic input trees having the same branching ratios. The building blocks in the 3rd layer of Fig 5 have additional inputs emanating from external nodes which prohibits these to be transformed into the Fibonacci base previously mentioned via a graph fibration. Surprisingly these have the same branching ratio, the external inputs only change the initial numbers in the sequence these produce.

Before all this, a stability analysis is conducted to determine the range at which the external stimuli do not induce any instabilities in these networks + ODE models.

**Simulation test 3 setup.**  We studied four networks and their binary/integer edge-weighted versions. $\tilde{A}_{ji}$ entries in Eq (10) were altered by subtracting a normally distributed value with increasing standard deviation and zero mean. Initial values were set to equilibrium/resting state to assess partitioning method reliability when edge weights are distorted. During our analyses, we kept the $\sigma$ of the *LoS* function equal to $0.1mV$.

## Simulation results

**Simulation test 1 results.**  Applying the *LoS* measurement to the last second of simulation time for each case produces blocks of synchronicity as seen in Figs 11 and 12. For all the 4 cases pertaining to the backward chemical network, all of the expected fiber partitionings are present (diagonal boxes delineated by red). We find that for the binary version of the backward chemical network, neurons within the same fiber became synchronous with nodes on other fibers, as an example, the 1*st* and 6*th* of these diagonal blocks starting from the top in Fig 11. We confirmed that these synchronizations are still present for longer simulation times or with a more restrictive $\sigma$ parameter. These were only confirmed to be significantly diminished when the networks were driven through an external sinusoidal or Gaussian random walk stimulus, as shown in the simulation test 2.

Results for the forward chemical model were similar and can be seen in Fig 13. The forward chemical networks synchronize to the predictions made by the fiber and orbit coloring. In the binary Chem type II case inter-neurons PVC and motor-neurons VB07 and VB09 synchronize as tested by more sensitive simulations.

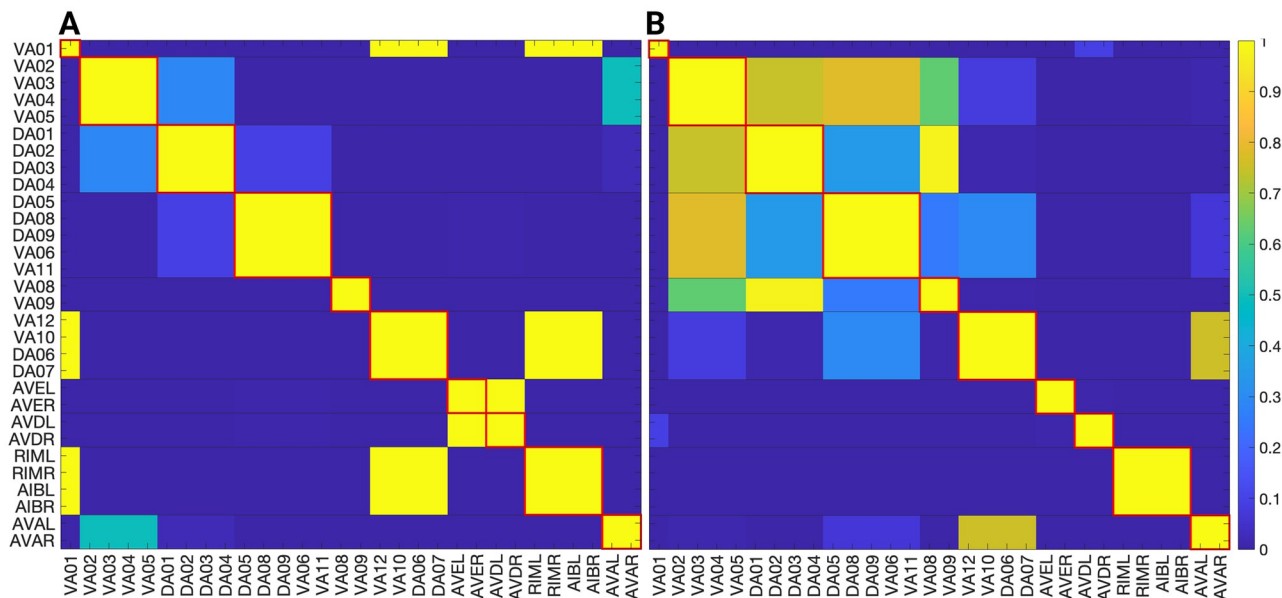

**Fig 11. Backward chemical model type I simulations with no external stimuli.** Initial voltages set to $-35mV$. (A) Left side pertains to the binary version of this symmetric network where an example of two synchronous fibers can be observed (AVDL+AVDR with AVEL+AVER) besides another example with 3 synchronous fibers. An equitable partitioning alone can not predict which fibers become synchronous with one another. (B) the right is the weighted version of this system which shows a perfect separation of the values of the nodes into the expected those from fiber partitioning. The block system shows the result of the *LoS* measurement done on the last second of the simulations. The red-lined boxes are visual indicators for the expected cluster synchronization of neurons predicted by the fiberation partitioning method.

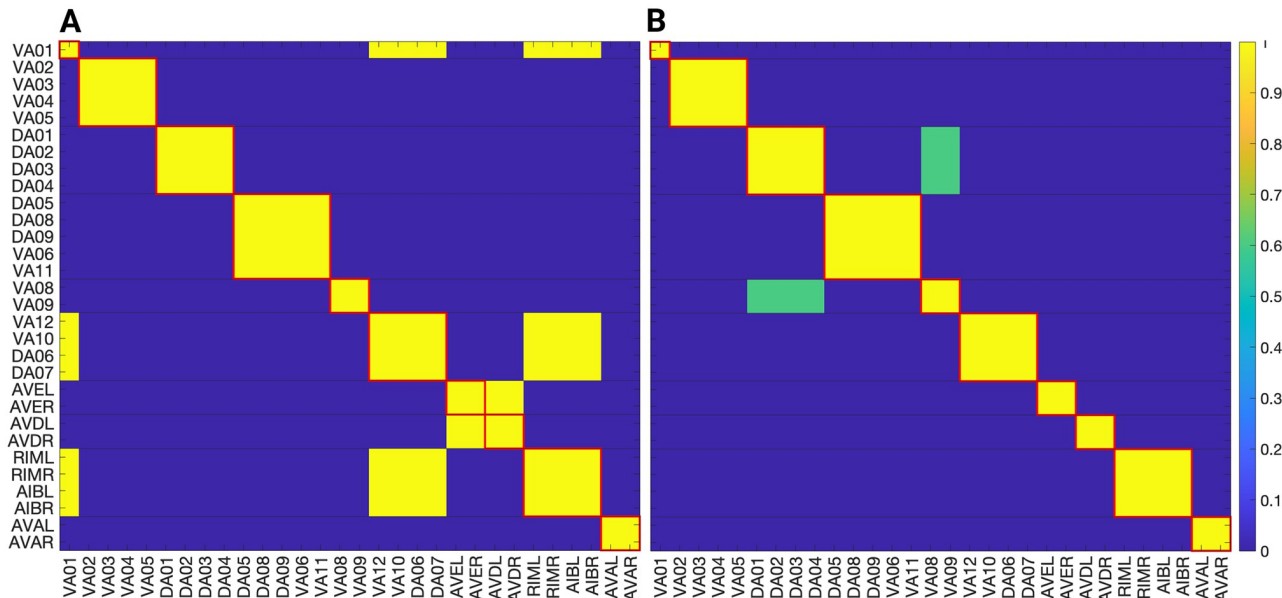

**Fig 12. Backward chemical model type II simulations with no external stimuli.** Synaptic variables initially set to $s_{eq}$ and initial voltages set to $-35mV$. (A) Left side pertains to the binary version of this symmetric network. (B) is the weighted version of this system which shows a perfect separation of the values of the nodes into the expected those from fiber partitioning. The block system shows the result of the $LoS$ measurement done on the last second of the simulations. The red-lined boxes are visual indicators for the expected cluster synchronization of neurons predicted by the fiberation partitioning method.

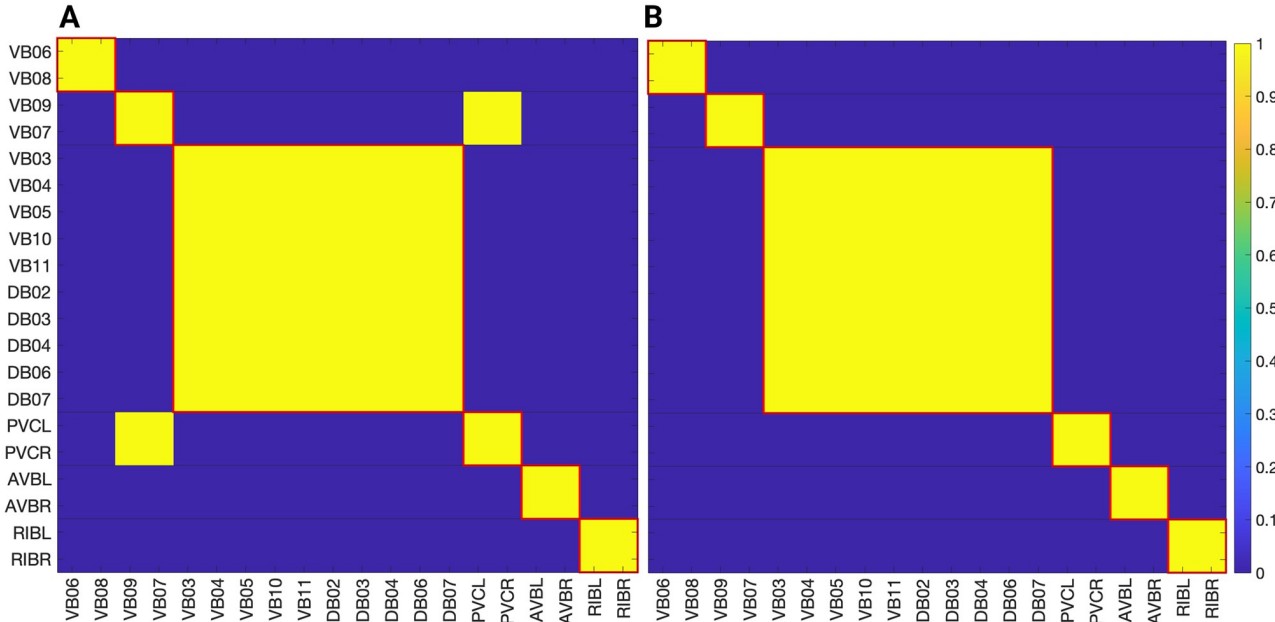

**Fig 13. Forward chemical type II model simulations with no external stimuli.** Synaptic variables initially centered around $s_{eq}$ with a standard deviation of 0.1 and initial voltages set to a mean of $-35mV$ with a $10mV$ standard deviation. (A) The left portion pertains to the network with binary edges. (B) The right portion is associated to the integer weight case. The block system shows the result of the $LoS$ measurement done on the last second of the simulations. The right $LoS$ matrix is an example of an ideal case. The red-lined boxes are visual indicators for the cluster synchronization of neurons under their expected balanced coloring partitioning.

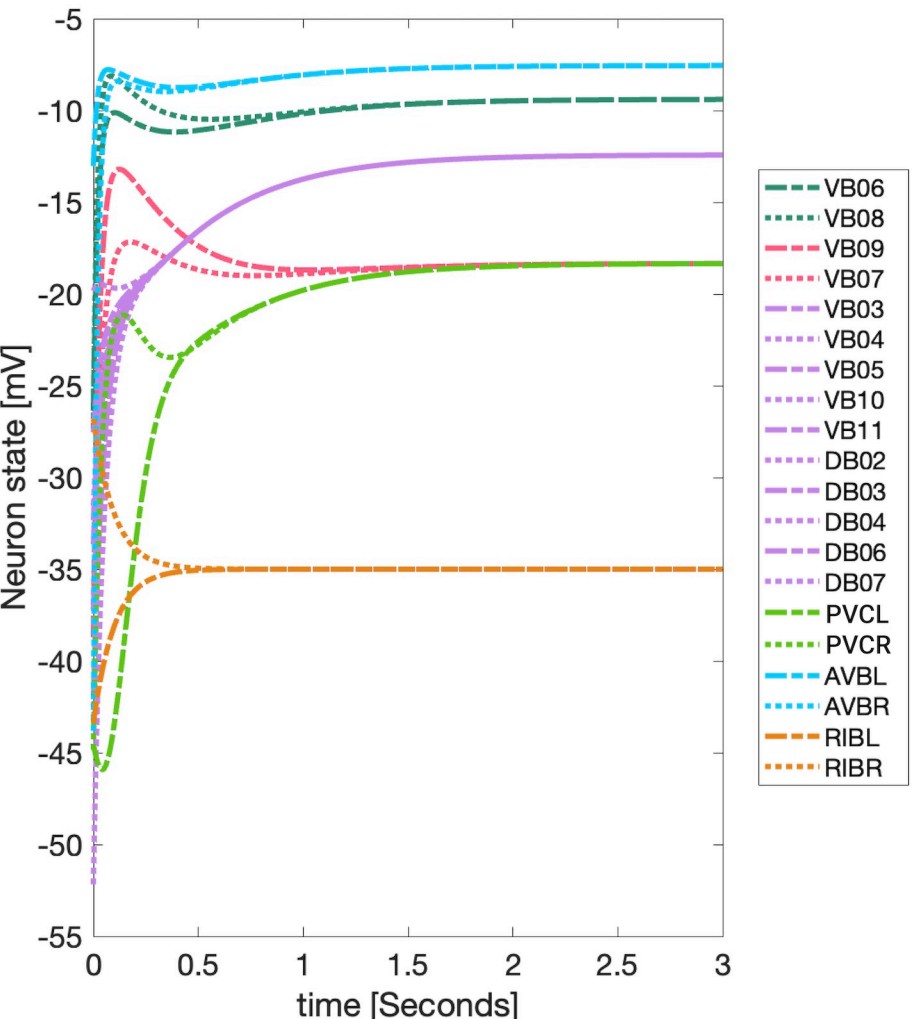

**Fig 14. Binary forward chemical type II model simulations with no external stimuli.** Synaptic variables initially centered around $s_{eq}$ with a standard deviation of 0.1 and initial voltages set to a mean of $-35mV$ with a $10mV$ standard deviation. Lines are color coded based on their fiber partitioning where it can be seen that all same colored neurons eventually synchronize with each other. Neurons PVC belonging to one fiber become synchronous before synchronizing with neurons VA08 and VA09 belonging to another fiber. The *LoS* associated with this simulation can be observed in Fig 13A where the latter example is captured by the only off diagonal blocks.

In Fig 14 it can be observed how these neurons belonging to different fibers become synchronous. Curiously one can notice that PVC neuron synchronize with each other and so do neuron VB07 and VB09 with each other before these two fibers synchronize.

**Simulation test 2 results.**   In this section, we aim to investigate the predictive power of fiber and orbit partitionings by simulating neural activity. To achieve this, we drive the four networks by delivering the same stimulus through left-right inter-neuron pairs. These left-right inter-neuron pairs belong to the same partition in all four networks, therefore not breaking the fiber partitioning.

The external stimuli in any of these networks ultimately changes the equilibrium solution of its respective ODE due the nature of these equations. The sinusoidal and noise stimuli will lead the voltages of the neurons away and towards equilibrium periodically and aperiodically

respectively. By such these two stimuli can be regarded as perturbations if kept relatively small compared to the constant term stimuli of Eq (17) which dictates the equilibrium point solutions. Additionally, it is important to ensure that the external stimuli do not induce any instabilities that could complicate the interpretation of results such as bifurcations or chaos which a fiber analysis of a graph will not be able to predict. Therefore, we conduct a stability analysis of the ODEs to determine the range within which the external stimuli does not cause any instabilities. As a common practice in the study of dynamical systems, we focus on the Jacobian of the model under study. For the Gap junction it's Jacobian can be written as:

$$\mathbf{J}_{Gap} = -\alpha^{leak}\mathbf{I} - \alpha^{gap}\mathbf{L} \tag{23}$$

where $\mathbf{I}$ is the identity matrix and $\mathbf{L}$ is the Laplacian of the gap junction adjacency matrix. Notice that Eq (23) is independent of any external stimuli. The Jacobian matrix is a useful tool for investigating the stability of a system's equilibrium point. By analyzing the eigenvalues of the Jacobian matrix, we can gain insight into the local behavior of the system near it's equilibrium point. According to the established principles of linear stability analysis, if all eigenvalues are real and negative, then the system is stable, meaning that any small perturbations around the equilibrium point will dampen over time, and the system will return to its stable state. If one or more eigenvalues have positive real parts, then the system is unstable, and any small perturbations will cause the system to diverge from the equilibrium point [96, 97].

We find that all eigenvalues of Eq (23) are negative for the values of $\alpha^{leak}$ and $\alpha^{gap}$ as described in the Ordinary differential equation parameters and implementation methods section. Therefore the gap junction model is stable under any external stimuli as it does not depend on $I^{ext}$.

Moving on, the individual terms of the Jacobian for the Chem type I model takes the form of Eq (24). At equilibrium, the sigmoid functions have a value of 0.5 (as per how $V^{threshold}$ is constructed in this model [85]). The Jacobian for this chemical model depends on the external stimuli indirectly through $V_i$ (Eq (20)) situated in the off-diagonal terms of the Jacobian.

$$\mathbf{J}_{ChemI} : \frac{\partial V_i}{\partial V_j} = -\alpha^{leak}\delta_{ij} - \alpha^{chem}\delta_{ij}\sum_{j=1}^{n}\tilde{A}_{ji}^{chem}\Phi(V_j)$$

$$+ \alpha^{chem}\tilde{A}_{ji}^{chem}\Phi(V_j)(1 - \Phi(V_j))\gamma(V_i - V_{s,j})(1 - \delta_{ij}) \tag{24}$$

Analyzing the eigenvalues of the Jacobian matrix at the stable point solutions when $V_i = V_i^{threshold}$, determined by Eq (18), shows which neurons will have unstable voltages for a given constant $I_j^{ext}$ value. For the F-Chem networks with integer weights under the Chem type I model, most of these eigenvalues remain real and negative as the external stimuli applied to neurons PVC or AVB are varied from $-500pA$ to $500pA$. An exception occurs for the eigenvalues associated with the neurons that receive external stimuli. For PVC stimulation, the real part of its eigenvalue linearly crosses into the positive regime for external stimuli greater than $3.08pA$ or less than $-2.38pA$ (Fig 15A). This eigenvalue is accompanied by another that mirrors its behavior. These two eigenvalues have the same value at $0.35pA$ of external stimuli (dashed black line in Fig 15). This occurs because when the external stimuli take this value, the leaking potential term in Eqs (12) and (13) approach zero. Simulations of this model for the respective network show a bifurcation between the left and right PVC inter-neurons values at values higher than $3.08pA$ of external stimuli. A similar result can be observed for the stimulating neurons AVA in the backward chemical network with binary weights as seen in Fig 15B. No simulation with negative external stimuli produced bifurcations.

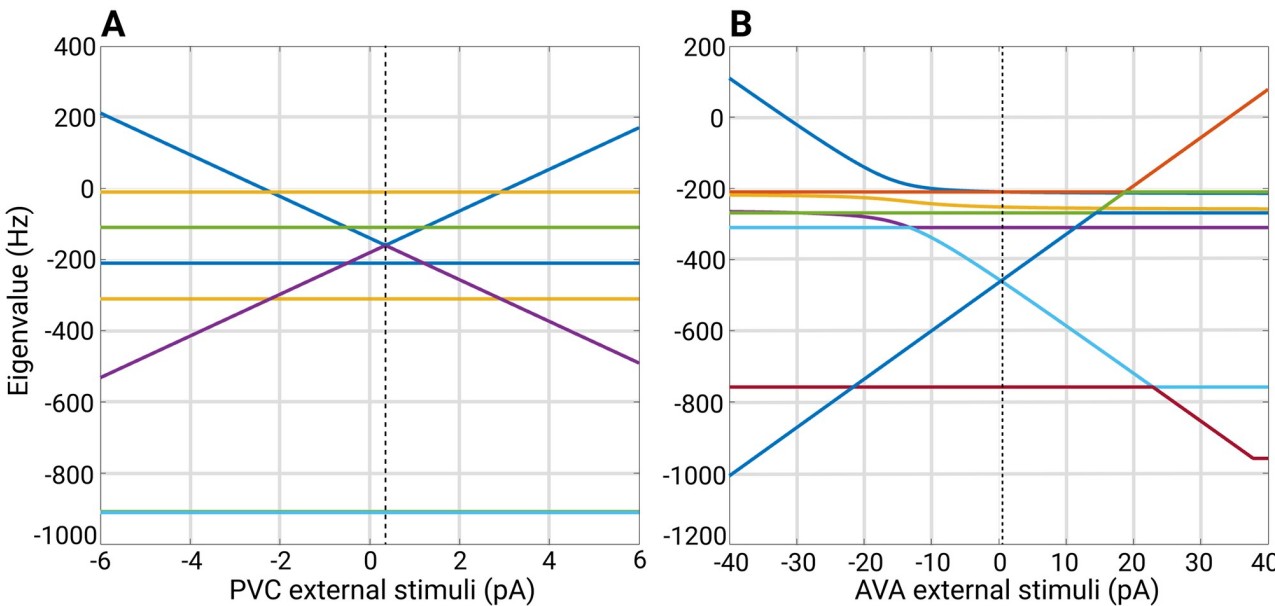

**Fig 15. Stability analysis results.** (A) Eigenvalues of the Jacobian for the integer edge weighted F-Chem network under the Chem type I model as the external stimuli through inter-neurons PVC is incremented from $-6pA$ to $6pA$. The first positive eigenvalue happens at $3.08pA$ of external stimuli. (B) Eigenvalues of the Jacobian for the integer edge weighted B-Chem network under the Chem type I model as a function of constant external stimuli input through AVA inter-neurons. First positive eigenvalue happens at $34.1pA$ of external stimuli.

A look into the Jacobian of the Chem type II model Eq (25) also indicates that it is stable under external stimuli which do not exceed specific values. Below one can find its Jacobian.

$$\mathbf{J}_{ChemII} = \begin{bmatrix} \dfrac{\partial \mathbf{V}}{\partial V} & \dfrac{\partial \mathbf{V}}{\partial s} \\[2ex] \dfrac{\partial \mathbf{S}}{\partial V} & \dfrac{\partial \mathbf{S}}{\partial s} \end{bmatrix}, \ \mathbf{V} = \{\dot{V}_i\}, \ \mathbf{S} = \{\dot{s}_i\}. \tag{25}$$

At equilibrium, the voltage values of the neurons $V_i$ equal that of the $V^{threshold}$ as in Eq (18) [85]. Therefore, the sigmoid function takes a value of 0.5, and the synaptic variable term equals $s_{eq}$. This Jacobian is indirectly affected by the external stimuli through the $V_i$ terms, which only appear in the off-diagonal term $\frac{\partial \mathbf{V}}{\partial s}$ as it can be seen in Eq (26).

$$\frac{\partial \mathbf{V_i}}{\partial s_j} = -\alpha^{chem} \tilde{A}_{ji}^{chem} V_i (1 - \delta_{ij}) \tag{26}$$

This and other stability analysis were carried out for all the combinations of ODE models, networks, and edge types with external stimuli varying from $0pA$ to $500pA$. In Table 1, we show the results where the values of this table indicate the strength of the external stimuli (into the indicated inter-neuron pairs) at which instability and the first positive eigenvalue arise. For all cases, only one positive eigenvalue (value underscored in Table 1) was above a reasonable strength based on multiple electrophysiological studies done on *C. elegans* [45, 46, 98].

Knowing the maximum strength of the external stimuli at which no instabilities appear, we stimulate their respective network + ODE model with a constant input equal to 90% of the external stimuli necessary to cause instability in the network plus a sinusoidal (or noise) stimulus with an amplitude 5% of the instability strength. To test through simulations that the

**Table 1. Instability analysis.** Results for all combinations of chemical ODE models, networks and edge types. An external stimuli drove each of these networks through increasing strength from 0 up to $500pA$ until a positive eigenvalue emerged. These neurons are shown in the third column. Results are given in pico-Amperes.

| ODE MODEL | NETWORK | NEURONS | BINARY($pA$) | INTEGER($pA$) |
|---|---|---|---|---|
| CHEM TYPE I | Forward | PVC | 1.5 | 3.08 |
|  |  | AVB | 14.46 | 265.4 |
|  | Backward | AVA | 54.15 | 34.1 |
|  |  | AIB+RIM | 2.29 | 2.29 |
|  |  | AVD | 11.31 | 14.83 |
| CHEM TYPE II | Forward | PVC | 1.06 | 1.25 |
|  |  | AVB | 58.72 | 4.51 |
|  | Backward | AVA | 13.6 | 8.51 |
|  |  | AIB+RIM | 1.12 | 1.12 |
|  |  | AVD | 4.89 | 3.46 |

neurons of these graphs synchronize to those stipulated by fiber and/or orbit partitioning we stimulate some of the neuron in these network. Gap junction networks were driven through inter-neurons AVB and motor-neuron DB04 for the forward network and inter-neurons AVA and motor-neuron DA05 for the backward network. This additional driving signal applied to a motor-neuron is needed for the Gap junction models as these networks globally synchronize if there is only one driving signal irrespective of the network. The introduction of a second signal with different characteristics (frequency and phase) aids in breaking the global synchronization revealing the expected cluster synchronizations. For the Chemical synapses networks we stimulate PVC neurons for the forward network and AVA neurons for the backward network. We set the inter-neurons frequency to 2Hz and that of the motor neurons to 1Hz. We calculated the *LoS* matrix for each of these simulations and rounded entries lower than 1 to 0.

These resulting matrices were subtracted from their idealized *LoS* matrix obtained from fiber partitioning where these have diagonal block elements equal 1 (elements inside squares delineated with red lines in Figs 11–13) and all other entries equal zero. This subtraction was divided by 2 times the number of elements in *LoS* matrix without counting diagonal elements. A zero value in these indicates a perfect agreement with the idealized *LoS* matrix from fiber partitioning. A negative value indicates additional synchronization between two or more balanced colored groups. A positive value would indicate that the expected fiber partitionings were not found or disagree in some form.

Table 2 presents the results of our analysis. According to these results, all networks, regardless of the ODE model or type of edge, separate into distinct synchronous groups under the

**Table 2. Synchronization among fibers.** Synchronization difference between the *LoS* matrix of a driven network and its ideal *LoS* matrix. A value of zero indicates perfect agreement between the *LoS* matrix of a driven networks and its ideal *LoS* matrix with distinct synchronous groups. A negative value indicates that two or more minimal balanced colorings have the same value at the same time. A positive value would indicate that the dynamics of its neurons do not cluster synchronize to the partitioning of the minimal balanced coloring. The networks here were driven through inter-neurons mentioned in Sections. The best ODE model to differentiate synchronous partitionings was the Chem type II model.

| ODE MODEL | NETWORK | BINARY | INTEGER |
|---|---|---|---|
| CHEM TYPE I | Forward | 0 | 0 |
|  | Backward | -0.009 | 0 |
| CHEM TYPE II | Forward | 0 | 0 |
|  | Backward | 0 | 0 |
| GAP TYPE | Forward | -0.12 | 0 |
|  | Backward | -0.16 | -0.16 |

*LoS* metric. This finding confirms the partitions obtained via fiber partitioning. We did not observe any agreement in orbit coloring for the B-Chem network, which would have led to a positive entry in Table 2 by dividing diagonal boxes into two or more diagonal boxes. The negative values in this table indicate that some of the cluster cells expected from fiber partitioning synchronize with one another, but this is not a negative result because fiber partitioning only requires neurons in a fiber to synchronize and does not preclude different fibers from synchronizing.

**Simulation test 3 results.**  We examined neuron synchronization and fiber partitioning agreement, testing robustness by adding random edge weights to simulate missing or uncertain information. We added normal distribution weights (std. dev. 0-0.1 in 0.01 steps, with 0 mean) to all weights, and stimulated the same neurons as in simulation test 2 in ten separate occasions with all the resulting *LoS* being averaged (see *LoS* matrices of Fig 16). The *LoS* obtained from simulation test 2 was binarized (elements below a value of 1 were zeroed) and used as a mask on the averaged results. The resulting matrices were subtracted from the masking matrix.

In this simulation, the driving signals are composed of a constant input of 0.1*pA* plus an oscillatory term with a frequency of 2*Hz* for inter-neurons and 1*Hz* for motor-neurons with an amplitude set to 0.5*pA* plus a Gaussian random walk scaled between −0.01*pA* and 0.01*pA*.

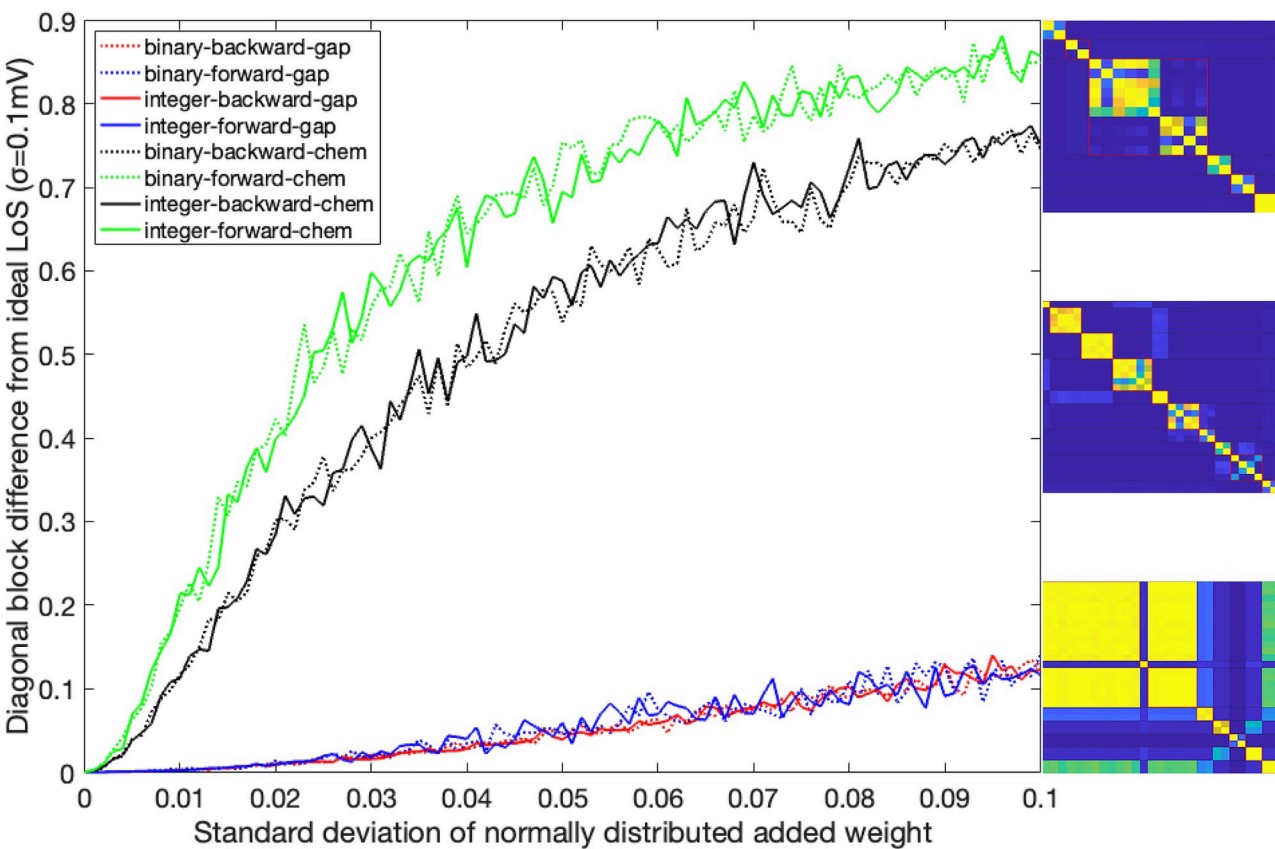

**Fig 16. Effects on synchronicity due to randomizing non-zero weights.** The addition of a normally distributed amount (with zero mean) to the non-zero weights of all the networks studied here (Gap and Chem type II models) is studied. Each perturbed network is simulated for 5 seconds with the last second used to calculate its *LoS* and subtracted from the idealized *LoS* (see Fig 13B for an example). The difference is only calculated on the expected minimal balanced coloring synchronizations (diagonal blocks). Examples of the *LoS* calculated for the F-Chem-Integer, B-Chem-Integer and B-Gap-Integer networks are shown from top to bottom respectively.

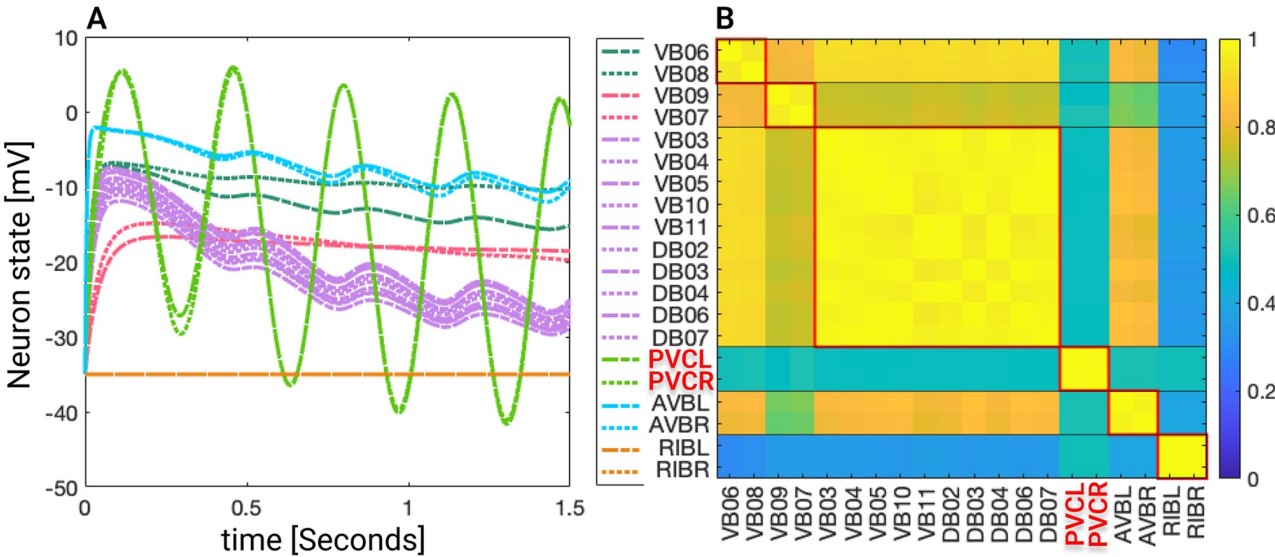

**Fig 17. Phase synchronization results.** (A) Dynamics for the forward chemical integer edge weighted network (with an average edge weight of 2.6 before any edge alterations). This network was driven through inter-neurons PVC as explained in section: Simulation test 3 results. The alterations to it's edges weights were done by adding to them normally distributed numbers with standard deviation 0.05 and mean 0. (B) Phase Locking Value (*PLV*) matrix of these dynamics.

This gives a max amplitude of $0.61pA$, below any of the values that would induce instabilities in any of our networks as seen in Table 2. All initial voltages were set to $-35mV$, and all initial synaptic variables to $s_{eq}$. Fig 16 shows the results. Varying weights affects *LoS* synchronicity of networks similarly, regardless of binary or integer edge weights (continuous and dotted lines, respectively). Backward chemical locomotion network is slightly more robust to weight changes than forward chemical locomotion network. All gap junction model synchronizations are robust to weight changes. However, most synchronicities were lost when measured through the strict *LoS* method, which requires signals to have the same value at the same time within a distance less than $\sigma = 0.1mV$. Different synchronicity measures may lead to different conclusions about the dynamics of stimulated networks. For instance, in the case of the integer edge-weighted F-Chem network with edge weights having a standard deviation of 0.05, its dynamics are illustrated in Fig 17. Although *LoS* analysis shows no synchronizations, the *PLV* metric demonstrates that neurons of the same minimal balanced coloring exhibit high synchronicity under phase synchronicity. As such, by relaxing the constraint of synchronicity from having the same value simultaneously to having the same instantaneous phase, it is possible to highlight the synchronicities expected from fibration symmetries. This was seen for all other networks as well.

## Discussion

We found that, in the case of the gap junction networks presented here, orbits and fibers coincide. This is a direct consequence of an undirected graph, where nodes have the same value for their in/out-degree. This similarity is lost when considering a direct graph where this condition is broken. Generally, in a directed graph, orbits and fibers do not coincide (unless some modular structures are present). As expected, we found that in these circumstances, the number of fibrations symmetries was higher than the automorphism symmetries (due to the less

restrictive constraints) for only one of the networks. More symmetries translate to less number of clusters, as depicted in Fig 4.

We discovered a non-trivial fiber partitioning of various neuronal classes. Across all subnetworks studied, we found that the left- and right members of each interneuron class AVA, AVE, AVD, AVB and RIB each receive a class-specific balanced coloring (Figs 4, 6 and 7). This is consistent with our previous findings that pairwise correlations between the activities of left-right- bilaterally symmetric neurons are the strongest across the worm brain [17]. Moreover, we find that motor-neurons in the ventral cord partition into unexpected sub-groups, that are not simply explained by their body positions along the anterior- to posterior body axis, nor by the body wall muscle segments they innervate (Figs 4, 6 and 7). These motor-neurons have yet not been recorded and identified altogether simultaneously, however our previous work indicates that all A-class and all B-class motor neurons are highly correlated with each other respectively when recorded in immobilised worms (reference. Kaplan, Salazar 2021). Our analyses performed here suggests, that those subgroups that receive the same balanced coloring synchronise further, a prediction that can be tested in future whole nervous system calcium imaging recordings. We speculate that these subgroups also relate to each other functionally, by perhaps contributing similarly to the worms locomotion behavior or controlling different aspects of it e.g. undulation speed, curvature or other aspects of body posture. Testing this hypothesis will require recording their activity simultaneously in freely crawling worms.

The fiber building blocks (FBB) of the directed networks with binary edges were explored. It was discovered that the B-Chem network is conformed by Fibonacci fibers which indicates that the inter-neurons conforming this network have nested loops which indicate a high degree of complexity for this group of nodes driving the motor-neurons of the system. For the F-Chem network did not show to be conformed by the FBB with nested loops but did show that some of its motor-neurons are composed by multilayered building blocks indicating that some of the neurons of this system become synchronous indirectly through neurons with two degrees of separation. This helps with the robustness of the network as destroying some neurons will not affect the synchronicity of other neurons in the same fiber. Our partitioning into elementary building blocks suggest that some of these motifs serve some computational roles in the generation of locomotion. Whole nervous system calcium imaging recording of all of these neurons simultaneously could provide insights into the types of these computations.

Understanding the stability of the equations used to simulate the neuronal interactions in these networks aided to determine the parameter space for which the partitions under different conditions were valid. The equitable partitions found by fibration and automorphism theory predicts only the existence of the cluster synchronization solutions and does not mention anything about their stability. Investigation of this aspect can only be done numerically. We performed simulations of interacting simulations according to Eq (10).

First, we assume no external stimuli, and we compare the partitions found by means of fibrations in the *C. elegans* network with those found through the Level of Synchronicity measure (*LoS*) based on dynamical simulations. For all networks and for all admissible ODE models regardless of the edge type we found perfect agreement between the expected minimal balanced coloring partitionings and the block of synchronicity. For the F-Chem type II weighted case, we found perfect separation of the expected fibers. In the binary case, we find that inter-neurons PVC and motor-neurons VB07 and VB09 synchronize. This deviation is more evident in the case of B-chem networks, where several minimally balanced colorings became synchronous with other minimal balanced colorings. Having two or more distinct clusters (a group of neurons with the same balance coloring) be synchronous with each other is not an unexpected result and can be expected based on the theory of equitable graph partitionings such as that or orbits and fibers. While the theory does stipulate that two nodes with

the same balanced coloring will be synchronous, it does not stipulate that two nodes with different balanced coloring will not be synchronous. There are no restrictions to how differently colored nodes behave [27, 99].

Overall, the integer-weighted version of the Chem network with synaptic variables is the best model when it comes to distinctively separating expected synchronous groups regardless of the ODE model. The final synchronization remained the same as in Fig 12 as the standard deviations for the initial voltages and synaptic variables varied from 0 to 0.1. This configuration is expected as the stable point solutions ($V^{threshold}$) of any of these models are independent of initial conditions [53, 85].

To simulate more realistic conditions, we increased the external stimuli, ensuring not to change the partitionings of the networks (balanced colored external stimuli). We found that the gap junction model is stable under any external stimuli as it does not depend on the external factor $I^{ext}$.

As for Chem networks, we found that at a strength greater than the ones indicated in Table 1, a voltage bifurcation arose for the pair of neurons with a $I^{ext}$ greater than zero and, by such, broke the pair's predicted synchronization from that of the minimal balanced coloring. The expected synchronization patterns of all other neurons remained the same as the ones predicted by the minimal equitable partitioning. All but one (underlined value) of the values in Table 1 seem within reasonably stable. This is consistent with electrophysiological recordings of a set of C. elegans neurons showing stable responses when stimulated with an external current source [14, 45, 98].

We found that integer-weighted chemical networks are more resistant to external stimuli when it comes to preserving their fiber structure. Their binary version behaves similarly, but for the case of B-chem, type II. No orbit coloring agreement was found for the B-Chem network. An agreement of this type would had lead to a positive value in Table 2, a result due to diagonal boxes being divided into two or more diagonal boxes (because we already know there are more orbits than fibers in this case, as noticed in the previous section).

We finally repeated the previous analyses in the scenario where only partial information about the connectome networks is available. According to our findings, missing information affects in a similar way both the binary and weighted versions of a given network. Nevertheless, gap-junctions networks are more robust to missing information, followed by backward chemical networks. Nevertheless, almost all synchronicities were lost when measured through the *LoS* method. This is due to the strong constraints required by the *LoS* metric. Indeed different metrics, like the *PLV*, reveal that neurons of the same minimal balanced coloring are still highly synchronous under phase synchronicity. These findings are also in par with [68] where a simple symmetric network of artificial neurons constructed from resistors and capacitor is still able to achieve synchronization for most configurations even though these components have an approximate 5% error in their expected parameters. The change in the networks weight can be transformed into each neuron having a different coupling strength ($\sigma^{Chem}$ or $\sigma^{Gap}$) where the networks weight are returned to their integer or binary setting.

The end goal of studying these sub-networks is to predict synchronization based on the connectivity patterns which based on theory these should be reflected as activity patterns in live recordings [17]. Indeed, Haspel *et al.* [100] created a repeating unit containing the different classes of motor-neurons and body-wall muscles that intend to capture missing links from [8]. Six of these units are stitched together in series to form the entirety of a symmetric neural infrastructure underlying the locomotion of the *C. elegans*; this inherently leads to the synchronicity of the neurons and muscles when modeled through simulations [101]. Similarly, we use symmetrized networks which are repaired versions of an original connectome that preserve

most of the original structure. On the repaired networks, we measure fibration symmetries to predict the synchronization of neurons based on their structure.

## Conclusion

Here, we present a theoretical framework suggesting that symmetries in the graph structure of a neuronal network underlie functional synchronizations. We explored some of the fiber building block of the networks presented here and found that they are composed by multilayered blocks or by block with nested loops. It was also shown that minimal balanced coloring could be successfully used to determine the synchronicity patterns of all the symmetrized networks. Its performance is best when only synaptic interactions are present compared to when only gap junction interactions are present mainly because zero to few different fiber become synchronous. Fiber and orbit symmetries were able to capture the synchronicity patterns that appear in purely gap junction networks. However, these synchronicity patterns are subtly distinguishable. All the undirected networks analyzed in this paper became globally synchronized when driven by constant input. Only when these gap junction networks were driven by sinusoidal and random stimuli did most of the predicted synchronicity groups recover. The simulation results hold true when compared to fiber coloring and should be a go to tool when creating symmetric networks for the *C. elegans* where simulation may be skipped as could have been the case in [61, 101].

On the contrary, orbit colorings do not perform as well as balance coloring in predicting synchronizations in ODEs models like Eq (10). If the system of coupled ODEs used accounted for the number of out going edges by varying some parameter, the dynamics of the neurons would possibly partition into the cluster predicted by orbit colorings. However, as of now, neurons and their potentials are known to only be affected by their inputs and not by the number of their synaptic outputs, therefore its reasonable to expect that the synchronizations of neurons in a connectome will obey fiber symmetries over orbit symmetries.

This study was performed under a set of idealised prerequisites i.e., a repaired connectome, nodes and edges with identical biophysical properties and isolated chemical- and gap junction networks. We, however, suggest that the network features described here significantly contribute to the activity patterns observable in the biological neuronal network of C. elegans. Our work makes testable predictions about neuronal ensembles that are expected to show elevated synchronicity and might group functionally in their control of locomotion. In future studies, we will test these predictions using experimental neuronal recording and circuit interrogation techniques. In the future, our approach can be applied to larger connectome datasets i.e., those of of larval or adult fruit flies [6, 102].

## Acknowledgments

Thanks to Wolfram Liebermeister for his suggestion to explore the effects of altering the weights of these networks. And thanks to Paolo Boldi for clearing out the air when it came to some equitable partitioning concepts.

## Author Contributions

**Conceptualization:** Manuel Zimmer, Hernán A. Makse.

**Funding acquisition:** Manuel Zimmer, Hernán A. Makse.

**Investigation:** Bryant Avila.

**Methodology:** Bryant Avila.

**Project administration:** Matteo Serafino.

**Software:** Bryant Avila.

**Validation:** Bryant Avila, Pedro Augusto.

**Visualization:** Bryant Avila.

**Writing – original draft:** Bryant Avila, Hernán A. Makse.

**Writing – review & editing:** Bryant Avila, Manuel Zimmer.

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
