## [Decision Letter · Decision Letter 0]

27 Sep 2023

PONE-D-23-15425Fibration symmetries and cluster synchronization in the *Caenorhabditis elegans* connectomePLOS ONE

Dear Dr. Avila,

Thank you for submitting your manuscript to PLOS ONE. We received one report from a referee, who rises a number of valid concerns. We invite you to submit a revised version of the manuscript that addresses the points raised during the review process.

We look forward to receiving your revised manuscript.

Kind regards,

Ivan Kryven

Academic Editor

PLOS ONE

Journal Requirements:

"Funding leading to the results of this work was provided to both H.M. and M.Z. by The National Institute of Biomedical Imaging and Bioengineering and National Institute of Mental Health through 1023

the National Institute of Health BRAIN Initiative Grant R01 EB028157.

https://www.nibib.nih.gov/

https://www.nimh.nih.gov/

https://braininitiative.nih.gov/

M.Z. is supported by the Simons 1024

Foundation (543069). The Research Institute of Molecular Pathology is funded by Boehringer Ingelheim.

https://www.simonsfoundation.org/

https://www.boehringer-ingelheim.com/

Reviewers' comments:

Reviewer's Responses to Questions

**Comments to the Author**

1. Is the manuscript technically sound, and do the data support the conclusions?

Reviewer #1: Partly

2. Has the statistical analysis been performed appropriately and rigorously? 

Reviewer #1: No

3. Have the authors made all data underlying the findings in their manuscript fully available?

Reviewer #1: No

4. Is the manuscript presented in an intelligible fashion and written in standard English?

Reviewer #1: Yes

5. Review Comments to the Author

Reviewer #1: This paper discusses application of tools from automorphism groups/symmetries and fibration symmetries to understand synchronization patterns in the locomotion neural sub-networks implicated in the forward and backward gait of the C. Elegans. I found the paper overall quite confusing.

I list below all of my questions/concerns.

Line 50, I could not understand this sentence "Where this synchronization is accomplished

when two or more neurons share a set of neurons as their inputs and where these

inputs have equivalent connectivity structures; such synchronization is explained

further in the paper"

Line 74, what does it mean that biological networks have mild constraints?

Line 117, I don't understand the relevance of studying chemical gap junctions and electrical synapses "in isolation". Wouldn't the patterns of synchronization observed in reality depend on both? This is a particularly serious criticism considering that the paper is trying to apply concepts from physics and mathematics to better understand biology

Line 245, A graph is defined in terms of two sets, the sets of nodes and the sets of edges, but here there are two distinct sets of edges: chemical and electrical, so a proper definition would be G(N,E_gap,E_synapse)

Line 301: Is the minimal graph fibration the same as the minimal balanced coloring? If yes, I would make it explicit

Line 308 "Nodes in the same fiber synchronize" however both physically and biologically this is only true if the synchronous solution is stable

Line 311: "it is possible for two or more fibers to synchronize" -- this should not be possible unless the synchronous solution is an equilibrium

Line 314 "We find the fiber partitioning algorithmically by initially coloring all nodes and

arrows with a unique color. After that, the algorithm recolors all nodes with the same

number of colored inputs with a new color, including their outgoing arrows. These

procedures continue until no recoloring is possible." -- this sounds very much like the algorithm proposed in "Chaos

. 2011 Mar;21(1):016106. doi: 10.1063/1.3563581." This algorithm produces the minimum balanced coloring very efficiently.

Section Building blocks -- I found this section very difficult to follow. I encourage the authors to rewrite this section by making sure that it is accessible to non-experts

Eq. (9) The external current term effectively modifies the symmetries and fibrations of the networks, unless currents are always pcked to be the same for nodes in the same orbit/fiber. This should be discussed

Eq. (16) makes it clear that the external current includes a noisy term. I think this deserves very careful consideration and discussion within the manuscript. This noisy term should act differently from the other terms in Eq. (16), since in practice noise will never be the same at different neurons. It should be clarified what is the correlation between noise terms at different nodes/neurons.

Line 541: "Noise dynamics were implemented through a modified stochastic Runge-Kutta method as proposed in [87], where the update time step for the noise is dt1/2." I found this discussion generic. I think it would be better to add a detailed description of the method in an appendix/supplement

Line 562: "To apply the LoS metric to our simulations, we allow each network to reach a

stable state after the initial transients by running it for a sufficient amount of time." -- what if the network does not reach a stable state after the initial transient? By stable state do the authors mean an equilibrium?

Line 591 I found this confusing "The resulting partitionings and their cluster cells are the same for both binary and

integer weighted networks since the graph structure G is the same regardless of edge

type. However, the input tree for a fiber associated with a specific neuron is likely

different between binary and weighted versions." I thought weights would play a role? If that's not the case, why is that?

Line 629 Obit-> Orbit

I found the sentences starting at lines 689 and 692 to contrast each other. How are the initial voltages set?

Same paragraph -- what is the standard deviation of the synaptic variable? Is this the noise term?

Line 700: syntactic variable?

Line 723: "we drive the four networks

by delivering the same stimulus through left-right inter-neuron pairs" what is the relevance of this? Are left-right inter-neuron pairs always in the same orbit or same fiber? The authors should distinguish perturbations that drive the system away from the synchronous solution, from those that do not.

Line 740: "ensure that the external stimuli do not induce any

instabilities that could complicate the interpretation of results" -- what does this mean and how does this lead to the Jacobian analysis?

Line750: "As a rule of thumb, if all

eigenvalues are real and negative, then the system is stable, meaning that any small

perturbations from the equilibrium point will dampen over time, and the system will

return to its steady state." That is not a rule of thumb. Also does this mean we are studying stability of fixed points? What happens when the external current is sinusoidal? What about the noise term?

Addititonally, I didn't understand the relevance/differences between the three simulation tests proposed. This point requires in my opinion careful consideration and a clearer discussion.

6. PLOS authors have the option to publish the peer review history of their article (what does this mean?). If published, this will include your full peer review and any attached files.

Reviewer #1: No

---

## [Author Response · Author response to Decision Letter 0]

26 Nov 2023

We have uploaded a pdf addressing all of the reviewer's comments, please find therein our responses.

Additionally, we have implemented all of the editor's comments, please inspect our revised manuscript and figures. If there is anything out of standard, please do not hesitate to let us know.

---

## [Editor Report · Decision Letter 1]

11 Jan 2024

Fibration symmetries and cluster synchronization in the *Caenorhabditis elegans* connectome

PONE-D-23-15425R1

Dear Dr. Avila,

We’re pleased to inform you that your manuscript has been judged scientifically suitable for publication and will be formally accepted for publication once it meets all outstanding technical requirements.

Kind regards,

Ivan Kryven

Academic Editor

PLOS ONE
---

## [Editor Report · Acceptance letter]

26 Jan 2024

PONE-D-23-15425R1 

PLOS ONE

Dear Dr. Avila, 

I'm pleased to inform you that your manuscript has been deemed suitable for publication in PLOS ONE. Congratulations! Your manuscript is now being handed over to our production team.

Kind regards, 

on behalf of

Dr. Ivan Kryven 

Academic Editor

PLOS ONE